# Scaling and quantization of large-scale foundation model enables resource-efficient predictions in network biology

Han Chen[1,2,3,4,12], Madhavan S. Venkatesh[1,2,5,12], Javier Gómez Ortega[1,2], Siddharth V. Mahesh[1,2,4], Tarak N. Nandi[6], Ravi K. Madduri ⬤ [6], Karin Pelka ⬤ [3,7,8,9,10,13] & Christina V. Theodoris ⬤ [1,2,4,11,13] ✉

Foundation models for network biology are pretrained on large-scale biological data to enable context-aware predictions in a diverse array of downstream tasks through transfer learning. However, increasing model sizes with the expansion of available pretraining data also increases the computational resources required for fine-tuning and inference in downstream applications. Here we first assemble a corpus comprising ~104 million human single-cell transcriptomes from a broad range of tissues and diseases and pretrain successively larger models, defining the scaling laws for transcriptional masked learning. We then demonstrate that model quantization preserves the contextual gene and cell embedding space of the full-precision model, matching performance in zero-shot and fine-tuning applications while requiring only 15% of the time and 34% of the memory as the full model for fine-tuning with the same batch size. Overall, model quantization represents an effective method for resource-efficient fine-tuning and inference while preserving biological knowledge.

Mapping gene regulatory networks in development and disease enables the discovery of key network regulators and network-correcting therapies that restore disease-dependent networks back to the normal state[1,2]. However, mapping the gene network architecture using traditional methods requires large amounts of transcriptomic data to learn the connections between genes, impeding discoveries in settings with limited data, including rare diseases and diseases affecting clinically inaccessible tissues. Yet, advances in sequencing technologies have driven a rapid expansion in the amount of single-cell transcriptomic data available from tissues more broadly. Standard approaches using

task-specific data to train a computational model to make predictions in that particular task require retraining from scratch with new task-specific data for each new task, therefore not fully taking advantage of the broader available data. In contrast, the machine learning approach of transfer learning leverages large-scale general datasets to pretrain models to gain foundational knowledge that can then be transferred to a vast array of downstream tasks, enabling predictions with little or no task-specific training data[3–5].

We previously developed a transfer learning strategy for network biology, pretraining a foundational deep learning model, Geneformer,

[1]Gladstone Institute of Cardiovascular Disease, San Francisco, CA, USA. [2]Gladstone Institute of Data Science and Biotechnology, San Francisco, CA, USA. [3]Gladstone-University of California, San Francisco (UCSF) Institute of Genomic Immunology, San Francisco, CA, USA. [4]Biological and Medical Informatics Graduate Program, UCSF, San Francisco, CA, USA. [5]Department of Computational and Systems Biology, University of California, Los Angeles, Los Angeles, CA, USA. [6]Data Science and Learning, Argonne National Laboratory, Lemont, IL, USA. [7]Department of Microbiology and Immunology, UCSF, San Francisco, CA, USA. [8]Helen Diller Family Comprehensive Cancer Center, UCSF, San Francisco, CA, USA. [9]Parker Institute for Cancer Immunotherapy, San Francisco, CA, USA. [10]Weill Cancer Hub West, San Francisco, CA, USA. [11]Department of Pediatrics, Institute for Human Genetics, Cardiovascular Research Institute, UCSF, San Francisco, CA, USA. [12]These authors contributed equally: Han Chen, Madhavan S. Venkatesh. [13]These authors jointly supervised this work: Karin Pelka, Christina V. Theodoris. ✉e-mail: christina.theodoris@gladstone.ucsf.edu

on ~30 million single-cell transcriptomes to enable predictions with limited data in network dynamics[6]. We demonstrated that this approach was able to drive biological insights that were experimentally verified with functional assays in preclinical models. For example, Geneformer discovered a novel transcription factor in cardiomyocytes with zero-shot learning and predicted candidate therapeutic targets for cardiomyopathy that improved contractility in an induced pluripotent stem cell (iPSC) model of the disease.

Overall, there has been a recent growth in the adoption of transfer learning for network biology, and multiple foundation models (scBERT, tGPT, scGPT, scFoundation, GeneCompass, UCE, Nicheformer) have been pretrained using large-scale single-cell transcriptional data to enable predictions in a diverse array of downstream tasks[6–13]. We previously demonstrated that larger and more diverse pretraining corpuses consistently boost predictions in downstream tasks[6], and as the data available for pretraining expand, larger models may enable zero-shot predictions in previously elusive settings. However, increasing model parameters also increases the computational resources required for fine-tuning and inference, increasing costs and limiting the accessibility of these biological foundation models in settings with low graphics processing unit (GPU) resources, including most academic research settings worldwide.

Here we demonstrate that model quantization enables resource-efficient fine-tuning and inference while retaining biological knowledge of the full-precision model. We first assembled a large-scale pretraining corpus, Genecorpus-104M, comprising ~104 million human single-cell transcriptomes from a diverse range of tissues and disease states from publicly available data. Pretraining Geneformer models with an expanded input size and a range of increasing parameters defined the scaling laws of transcriptional masked learning and found that larger models learned faster per token of data observed, analogous to foundation models in other fields[14]. While the larger models required greater computational resources, model quantization was able to match the performance of the full-precision models in zero-shot, few-shot and fine-tuning tasks and preserve context-specific representations in the gene and cell embedding space. Quantized models required only 15% of the time and 34% of the memory as the full-precision model for fine-tuning with the same batch size, with even greater practical time gains given that batch sizes could be increased due to the reduced memory requirements. Overall, model quantization represents an effective strategy for resource-efficient fine-tuning and inference while retaining biological knowledge, enabling cost savings and expanding the accessibility of biological foundation models.

## Results

### Model performance scaled as a power law with increasing parameters

We previously reported that increasing the size and diversity of the pretraining corpus for Geneformer consistently improved the model's predictive potential[6]. Since the pretraining of Geneformer in June 2021, there has been a substantial expansion in both the amount and diversity of publicly available human single-cell transcriptomic data, suggesting

that we could use these data to now train an even more effective foundational model. Therefore, we expanded our corpus to ~104 million human single-cell transcriptomes from an even more diverse array of tissue and disease contexts (Fig. 1a,b, Supplementary Table 1 and Extended Data Fig. 1a–d).

We balanced the data such that no tissue composed more than 25% of the data and performed scalable quality-control filtering. We also performed deduplication of studies by DOI to preclude training with duplicated cells, which can substantially overestimate corpus size due to studies being deposited in multiple databases (Extended Data Fig. 1a). The pretraining corpus also excluded cells with high mutational burdens such as malignant cells and immortalized cell lines. We excluded these cells as the high mutational burden may involve gain of function variants that alter gene functions from what the model would interpret in other cells with low mutational burdens. Each cell transcriptome was then presented to the model as a rank value encoding, which is a non-parametric representation of the transcriptome where genes are ranked by their expression in that cell scaled by their non-zero median value of expression across the entire pretraining corpus, as previously described[6]. The scaling factor takes into account each gene's typical expression range so that genes are ranked by their relative overexpression versus repression within this range in the given single-cell transcriptome presented to the model. Rather than ranking by absolute expression value, which would generally prioritize ubiquitously highly expressed housekeeping genes, the scaling factor deprioritizes housekeeping genes by scaling them to a lower rank. Conversely, genes such as transcription factors that may be expressed at low levels when they are expressed but have a high dynamic range across distinct cell states will move to a higher rank within the encoding in the cells where they are expressed in the higher end of their given range (Extended Data Fig. 1d). Furthermore, the rank-based approach may be more robust against technical artifacts that may systematically bias the absolute transcript counts value whereas the overall relative ranking of genes within each cell remains more stable.

Pretraining on more diverse data from a subsample of the updated Genecorpus-104M improved pretraining loss compared with pretraining on an equivalent amount of data from the less diverse Genecorpus-30M assembled in 2021 (Fig. 1c). In addition to increased diversity, however, more recent data also now detect more genes per cell (a few thousand genes per cell within the model dictionary of 20,271 genes) due to advances in single-cell RNA sequencing (scRNA-seq) technology. While pretraining with an input size of 2,048 genes per cell fully encompassed 93% of cells in Genecorpus-30M assembled in 2021, current data require a larger input size to represent the larger number of genes detected per cell with the current technologies. As such, we also expanded the model's input size to 4,096 genes per cell, which fully encompassed 93% of the cells in the updated Genecorpus-104M (Fig. 1d). Owing to the quadratic time complexity of dense attention, this doubling of the input size increased the computational intensity quadratically, but allowed the model to learn from a larger gene network context for each cell. In total, the updated pretraining corpus comprised ~150 billion gene tokens within the dictionary of 20,271 genes.

**Fig. 1 | Geneformer model scaling laws. a**, Schematic of Geneformer transfer learning strategy. Large-scale self-supervised pretraining on a generalizable learning objective yields a pretrained model with foundational knowledge relevant to network dynamics. This knowledge can be democratized to a vast array of downstream applications either through zero-shot learning (using the pretrained model directly) or fine-tuning with limited data. **b**, Organ representation of Genecorpus-104M. **c**, Pretraining loss of Geneformer pretrained on either the initial Genecorpus-30M assembled in 2021 or on an equivalent number of tokens subsampled from the more diverse, updated Genecorpus-104M. In both cases, models were ~40 million parameter models of the same architecture with input size of 2,048. **d**, Number of detected genes

per cell in Genecorpus-104M. Input size of 4,096 fully encompasses 93% of cells. **e**, Pretraining loss per token for Geneformer models of increasing parameters. **f**, Pretraining loss per compute budget (floating-point operations (FLOPs)) for Geneformer models of increasing parameters. **g**, Evaluation loss on held-out cells for Geneformer models of increasing parameters. **h**, Percent of correct masked gene predictions per held-out cell for either baseline approach or Geneformer pretrained for 0.1 epochs with 10% of Genecorpus-104M. In each cell, 15% of genes are masked. The baseline approach predicts the gene that is most frequently in a given masked rank within the rank value encoding based on the distribution of that rank position in cells within the 10% of Genecorpus-104M used to pretrain the partially trained Geneformer.

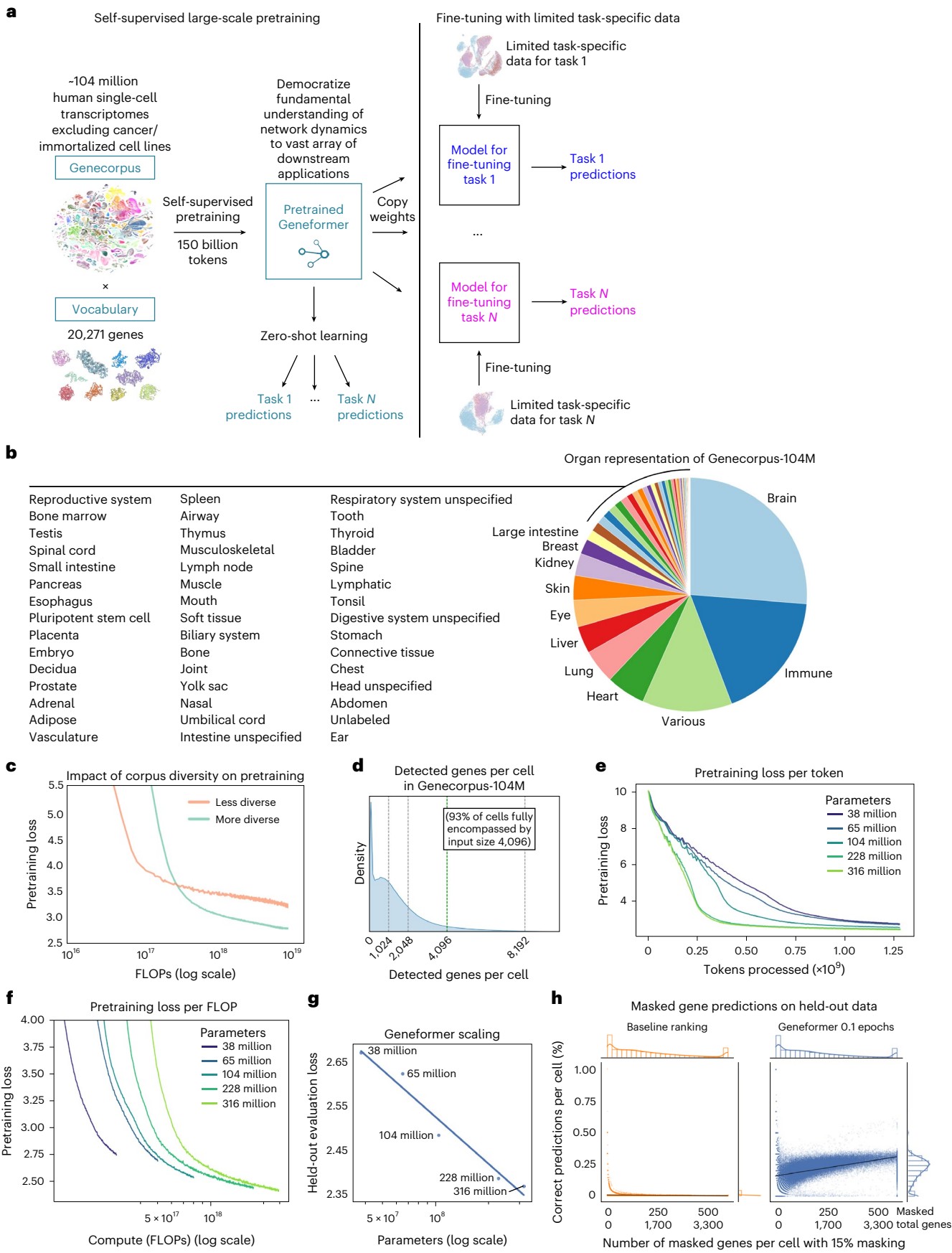

We then pretrained successively larger Geneformer models with the expanded input size to define the scaling laws of transcriptional masked learning. We found that models with a larger number of parameters scaled as a power law to improve held-out evaluation loss. Furthermore, we found that larger models learned faster per token of data observed, analogous to foundation models in other fields[14] (Fig. 1e–g).

### Zero-shot predictions in gene-level tasks improved with increasing model parameters

Geneformer pretraining is achieved with a self-supervised masked learning objective where the model learns to predict the identity of masked genes based on the context of unmasked genes within the gene network. The model thereby gains a generalizable understanding of gene network dynamics by observing how genes interact within a vast number of gene network states. Notably, predicting masked genes within a transcriptome is not the end goal of the model and is used only as the pretraining objective. Nevertheless, previous reports[15] have questioned whether single-cell foundation models are able to predict masked genes to a greater degree than baseline approaches such as predicting the gene most likely to occur in a given rank.

Foundation models pretrained with a masked learning objective are required to predict the exact gene from the 20,271 gene model vocabulary that should occur within a specific masked position in a given cell from the full diversity of tissues, developmental stages and disease states in the pretraining corpus. When pretrained on only 10% of the data, the Geneformer model significantly outperformed the baseline masked learning predictions on held-out data matched in diversity to the pretraining corpus (Fig. 1h). Geneformer prediction accuracy scaled with increasing context in cells with greater numbers of genes detected by scRNA-seq. Baseline predictions based on the most likely gene to occur in a given rank were generally unable to correctly predict the exact gene within an appreciable percentage of the masked positions in a given cell.

We next evaluated the robustness of the Geneformer contextual gene embedding space to batch-dependent artifacts that commonly affect scRNA-seq data. We found that embeddings of different genes were clearly separated in embedding space based on the high cosine similarity between embeddings of the same gene and the low cosine similarity between embeddings of different genes, with a high geometric separability index of 1.0. Embeddings of the same gene remained highly cosine similar in cells from different batches, demonstrating robustness to batch effects from the genome reference version, the Cell Ranger version for preprocessing, the cell preservation method and the sequencing platform, comparing single-cell with single-nucleus RNA-seq (Fig. 2a).

Then, we tested the zero-shot and few-shot performance of the intermediate (104 million parameters) and large (316 million parameters) Geneformer models on a diverse panel of biologically meaningful downstream tasks (Fig. 2b). We tested the models in the domains of distinguishing (1) disease genes (dosage-sensitive versus -insensitive transcription factors), (2) downstream targets of a transcription factor without perturbation data (NOTCH1 targets versus non-targets), (3) chromatin dynamics from transcriptomic data alone (bivalent versus Lys4-only methylated promoters), and (4) transcription factor regulatory range with no information of genomic distance (transcription factors that act in long versus short range with their targets). In this diverse set of tasks, the original Geneformer model pretrained in 2021 (GF-10M) was able to significantly improve predictions compared with alternative approaches with fine-tuning in all tasks (Fig. 2b). However, few-shot learning was unable to sufficiently train the model in tasks 3 and 4, which may be especially challenging to ascertain using only transcriptomic data as input with no information about epigenetic state or genomic distance.

Few-shot learning was sufficient for the intermediate model (GF-104M) to surpass full fine-tuning of alternative approaches in all tasks (Fig. 2b). Furthermore, the largest model (GF-316M) surpassed fully fine-tuned alternative approaches for all tasks with zero-shot learning, which is particularly valuable in settings without available task-specific data such as rare diseases or clinically inaccessible tissues (Fig. 2b,c). Overall, increasing model parameters improved zero-shot and few-shot predictions in gene-level downstream tasks.

### Model quantization allowed resource-efficient fine-tuning with nearly equivalent predictive accuracy

While increasing model parameters improved predictions, larger models required more computational resources for fine-tuning. Increased processing time leads to higher computing costs, and increased memory requirements limit accessibility to hardware with larger memory capacity. To address this, we implemented model quantization to 4-bit precision using quantized low rank adapters (QLoRA)[16]. This approach backpropagates gradients through the frozen, 4-bit quantized Geneformer into low-rank adapters to reduce memory usage and training time.

The 4-bit quantized Geneformer required only 15% of the fine-tuning time as the full-precision model with the same batch size (Fig. 2d). In addition, the quantized model required only 34% of the memory as the full-precision model with the same batch size (Fig. 2e). Of note, because the memory usage is lower, the true maximal time-savings are substantially larger as larger batches of data can be run through the 4-bit model at the same memory scale.

Despite the lower compute requirements, the 4-bit quantized Geneformer models matched the few-shot performance of the respective full-precision model (104 million or 316 million parameters) with nearly equivalent predictive accuracy in all gene-level tasks (no statistically significant change in performance; Fig. 2b). Overall, model quantization allowed resource-efficient fine-tuning, saving both time and memory, with no appreciable impact on predictive potential.

---

**Fig. 2 | Effect of scaling and quantization on gene-level tasks. a**, Effect on Geneformer gene embeddings from batch-dependent artifacts of genome reference (GRCh37 versus GRCh38), Cell Ranger (CR) version (2 versus 3 versus 7), preservation method (fresh versus frozen) and sequencing platform (single cell versus single nucleus). $n = 100$; the center box limits indicate the upper and lower quartiles. **b**, Zero-shot, few-shot or fine-tuned test set macro F1 of full-precision versus quantized models on diverse panel of biologically meaningful downstream tasks distinguishing disease genes (dosage-sensitive versus -insensitive transcription factors), downstream targets of a transcription factor without perturbation data (*NOTCH1* targets versus non-targets), chromatin dynamics from transcriptomic data alone (bivalent versus Lys4-only-methylated promoters) and transcription factor regulatory range with no information of genomic distance (transcription factors that act in long versus short range with their targets). $n = 3$ seeds; *$P = 0.049$, two-sided Wilcoxon rank sums versus best alternative; NS, non-significant; the center line indicates the median, the box

limits indicate the upper and lower quartiles, and the whiskers indicate 1.5× the interquartile range. GF, Geneformer (labeled by parameter count: 10 million, 104 million or 316 million); SVM, support vector machines; RF, random forest; LR, logistic regression; GF-10M, original Geneformer model pretrained in 2021; NA, not applicable. **c**, Zero-shot predictions by GF-316M in distinguishing bivalent versus Lys4-only-methylated promoters genome-wide. AUC, area under the receiver operating characteristic curve. **d**, Relative processing time (same batch size) for gene-level fine-tuning with 4-bit quantized versus full-precision GF-316M model. Time was quantified per same batch size, but quantized fine-tuning would take even less time in actuality because lower memory requirements allow larger batch sizes. **e**, Relative GPU memory requirements (same batch size) for gene-level fine-tuning with 4-bit quantized versus full-precision GF-316M model. In **d** and **e**, $n = 3$ trials; *$P = 0.049$, two-sided Wilcoxon rank sums; bar indicates the mean and error bars indicate the standard deviation; time/memory was equivalent across multiple trials.

## Zero-shot predictions in cell-level tasks improved with increasing model parameters

We next evaluated the influence of increasing model parameters on performance in cell-level tasks. We first evaluated the robustness of

the Geneformer contextual cell embedding space to batch-dependent artifacts. Geneformer cell embeddings were robust to batch effects from the genome reference version, the Cell Ranger version for pre-processing and the cell preservation method (Extended Data Fig. 2a,b).

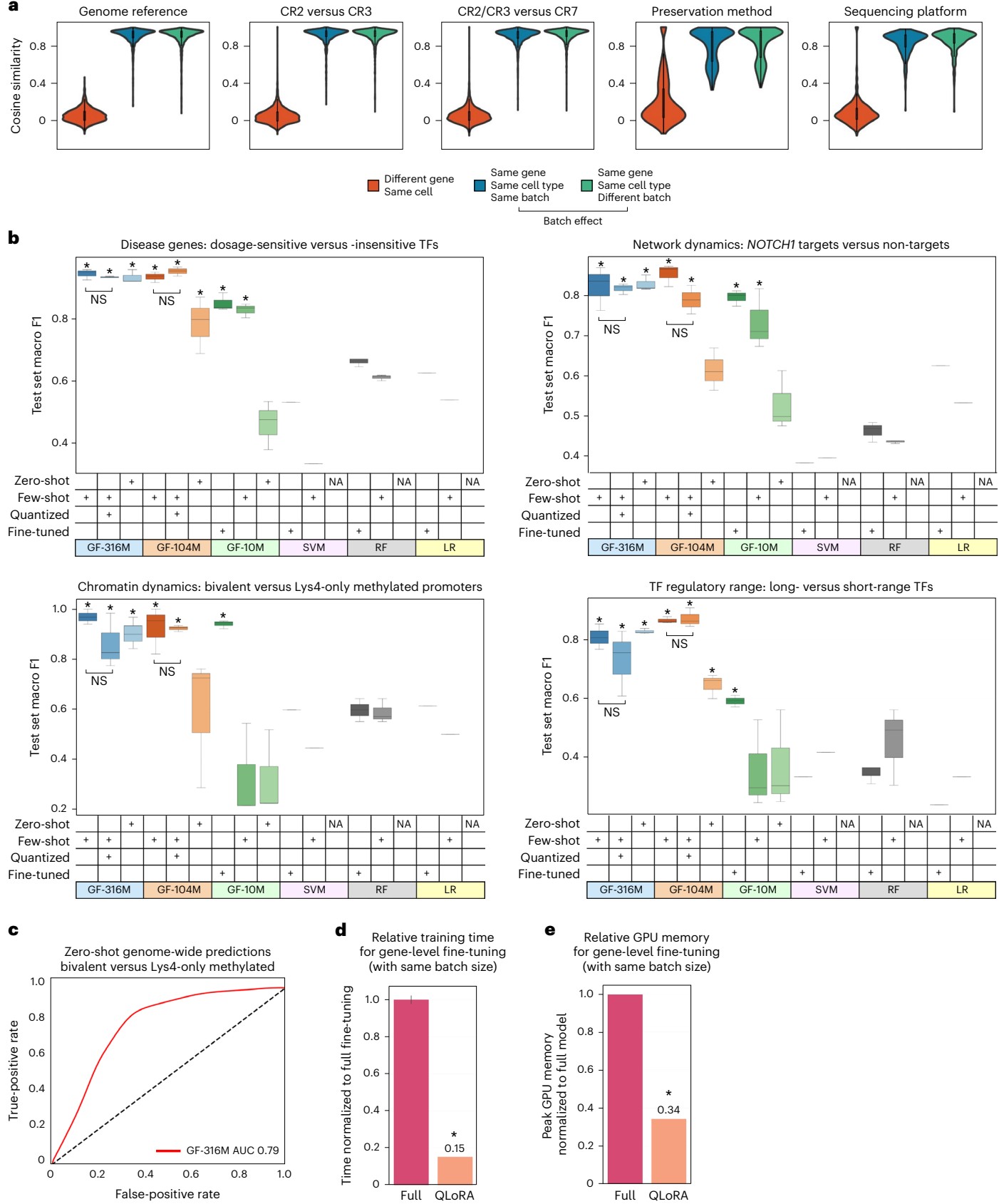

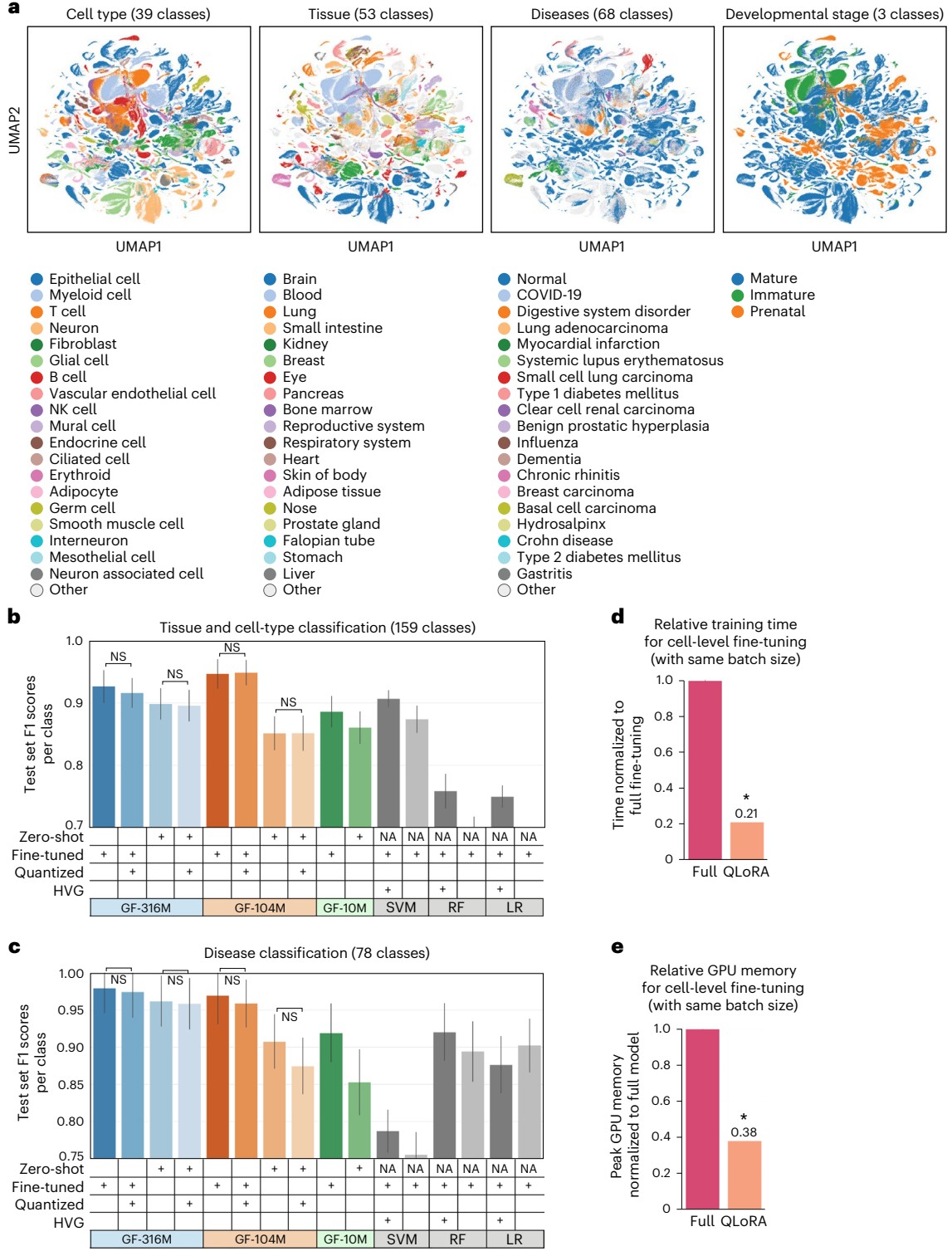

**Fig. 3 | Effect of scaling and quantization on cell-level tasks. a**, Zero-shot cell embeddings from GF-104M for 779,905 representative cells from the CELLxGENE corpus. **b**, Zero-shot or fine-tuned test set F1 score per class of full-precision versus quantized models in distinguishing cells by concatenated tissue and cell-type labels ($n = 159$ total classes). Median test set F1 of LR trained without feature selection was below the minimum $y$-axis limit. **c**, Zero-shot or fine-tuned test set F1 score per class of full-precision versus quantized models in distinguishing cells by disease ($n = 78$ total classes). **d**, Relative processing time (with same batch size) for cell-level fine-tuning with the 4-bit quantized versus full-precision GF-316M model. Of note, time was quantified per the same batch size, but quantized fine-tuning would take even less time in actuality because

the lower memory requirements of the model would allow larger batch sizes. **e**, Relative GPU memory requirements (with same batch size) for cell-level fine-tuning with the 4-bit quantized versus full-precision GF-316M model. In **b** and **c**, Geneformer models are labeled by parameter count (10 million, 104 million or 316 million). HVG, training on highly variable genes selected from training data as opposed to all genes; NS, non-significant difference between full-precision and quantized model, two-sided Wilcoxon rank sums; bars indicate the median and error bars indicate the standard error. In **d** and **e**, $n = 3$ trials, $*P = 0.049$, two-sided Wilcoxon rank sums, bars indicate the mean and error bars indicate the standard deviation; time and memory usage was equivalent across multiple trials.

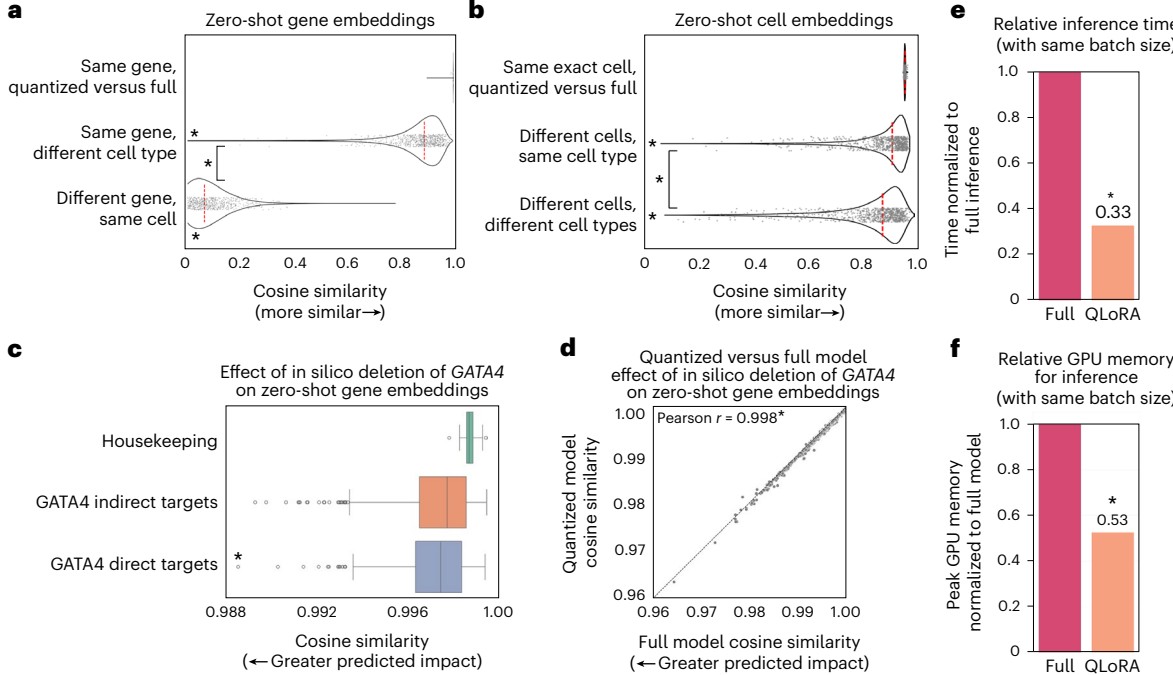

**Fig. 4 | Effect of quantization on the contextual embedding space and in silico perturbation. a**, Cosine similarity of zero-shot gene embeddings for the same gene in quantized versus full-precision models, same gene in a different cell type or different gene in the same cell. *$P$ = 0, two-sided Wilcoxon rank sums compared with quantized versus full-precision distribution except where indicated. **b**, Cosine similarity of zero-shot cell embeddings for the same exact cell in quantized versus full-precision models, different cells of the same cell type or different cells of different cell types. *$P$ = 0, two-sided Wilcoxon rank sums compared with quantized versus full-precision distribution except where indicated. **c**, In silico deletion of *GATA4* in iPSC cardiomyocytes[19] with GF-316M significantly more deleterious to GATA4 direct targets[20] than housekeeping genes or indirect targets. *$P$ = 0.000033, two-sided Wilcoxon rank sums; the

center line indicates the median, the box limits indicate the upper and lower quartiles, and the whiskers indicate 1.5× the interquartile range. **d**, Correlation of cosine shifts in response to in silico deletion of *GATA4* in the zero-shot quantized versus full-precision embedding space. *$P$ = 0, two-sided Pearson. **e**, Relative processing time (with same batch size) for inference with the quantized versus full-precision GF-316M model. Time was quantified per the same batch size, but quantized inference would take even less time in actuality because the lower memory requirements of the model would allow larger batch sizes. **f**, Relative GPU memory requirements (with same batch size) for inference with the quantized versus full-precision GF-316M model. In **e** and **f**, $n$ = 3 trials, *$P$ = 0.049, two-sided Wilcoxon rank sums, the bars indicate the mean and the error bars indicate the standard deviation; time and memory usage was equivalent across multiple trials.

Although gene embeddings were robust to batch effects from sequencing platform, the cell embeddings were more affected by the significantly lower number of genes detected by the single-nucleus RNA-sequencing compared with the same samples sequenced on the single-cell platform (Extended Data Fig. 2c–e). We tested whether fine-tuning the model to distinguish cell types using data sequenced on only one of the two platforms would sufficiently focus the model on biologically relevant features as opposed to batch artifacts. Indeed, fine-tuning only on single-cell RNA-seq data sufficiently oriented the embedding space such that unseen single-nucleus RNA-sequencing data accurately mapped onto the reference embedding space, primarily clustering by cell type rather than sequencing platform (Extended Data Fig. 2d,e). Of note, even the zero-shot Geneformer embedding space improved clustering by cell type compared with the original data, with or without batch correction by common methods ComBat[17] and Harmony[18], but fine-tuning further improved the batch integration within the embedding space (Extended Data Fig. 2d,e).

Overall, the zero-shot embedding space appeared to yield context-specific representations of cells separated by the biologically meaningful attributes of cell type, tissue, diseases and developmental stage (Fig. 3a and Extended Data Fig. 3a). However, although two-dimensional projections are useful for visualization, the separability of biologically meaningful classes is best evaluated in the full dimensional embedding space with quantitative measures such as testing the ability of a classifier to distinguish the relevant classes. Therefore, to quantify the separability of the embedding space, we tested whether cells could be distinguished into 159 classes representing concatenated

tissue and cell-type labels for non-diseased adult samples from CELLx-GENE (Fig. 3b). The zero-shot embedding space of the largest model (GF-316M) was the most separable by tissue and cell type compared with the zero-shot embedding spaces of smaller models and enabled more accurate predictions of tissue and cell type than the original data, even with highly variable gene selection. Similarly, the zero-shot embedding space of the largest model (GF-316M) was also the most separable by disease across 78 classes annotated in CELLxGENE (Fig. 3c). Fine-tuning the embedding space further improved tissue, cell-type and disease separability, also outperforming alternative models trained on raw counts or highly variable genes.

**Model quantization preserved the contextual gene and cell embedding space of the full-precision model**

We then tested whether the quantized embedding space would maintain the separability of tissues, cell types and diseases in both zero-shot and fine-tuned cell-state classification. Indeed, the quantized Geneformer models matched the predictive accuracy of the respective full-precision models in both zero-shot and fine-tuned settings. Zero-shot tissue/cell-type and disease classification by the quantized version of the largest model continued to outperform the zero-shot classification by smaller full-precision models, while offering significant advantages in resource efficiency by both training time and GPU memory usage (Fig. 3b–e).

When examining the cosine similarity of zero-shot gene embeddings in the quantized versus full-precision model, we found a negligible effect of quantization compared with the impact of the biological

context of that same gene's embedding in a different cell type, or a different gene's embedding (Fig. 4a). Similarly, quantization had negligible impact on the zero-shot cell embeddings compared with the impact of different cells or different cell types (Fig. 4b).

The preservation of biologically meaningful representations within the contextual gene and cell embedding spaces opens the opportunity to use the quantized models for large-scale in silico perturbation screens to markedly increase throughput and save computing costs. To confirm the ability of the quantized model to measure shifts in the embedding space due to perturbations, we tested in silico deletion of *GATA4*, a known congenital heart disease gene. In silico deletion of *GATA4* in iPSC-derived cardiomyocytes[19] with the full-precision GF-316M model had a significantly greater impact on known direct targets (as defined by chromatin immunoprecipitation sequencing (ChIP–seq)[20]) compared with indirect targets and housekeeping genes that it is not expected to regulate (Fig. 4c and Extended Data Fig. 3b). The embedding cosine shifts in response to in silico deletion of *GATA4* using the quantized model were highly correlated with the full-precision model, demonstrating that the quantized model is able to equivalently quantify shifts in the embedding space due to in silico perturbations (Fig. 4d).

As with fine-tuning, inference time and memory requirements were also significantly reduced by model quantization. The quantized model required only 33% of the inference time as the full-precision model with the same batch size (Fig. 4e). Furthermore, the quantized model required only 53% of the memory for inference as the full-precision model with the same batch size (Fig. 4f). The true maximal time savings are also substantially larger than what we report here as the reduced memory requirements enable larger batches of data to be run in parallel at the same memory scale.

Overall, model quantization enabled inference with significantly reduced time and memory requirements, increasing throughput and saving costs while preserving a biologically meaningful embedding space for modeling in silico perturbations.

## Discussion

In sum, we demonstrate that model quantization with low-rank adapters is an effective strategy for resource-efficient fine-tuning and inference while retaining predictive accuracy of full-precision foundation models for network biology. Geneformer models pretrained on ~150 billion gene tokens from >100 million single-cell transcriptomes scaled as a power law with increasing parameters, with the largest model showing the highest zero-shot performance on both gene- and cell-level biologically meaningful tasks. The 316 million parameter model pretrained on >100 million transcriptomes significantly outperformed the original 10 million parameter Geneformer model pretrained in 2021 on ~30 million transcriptomes. The updated Genecorpus, Geneformer model and code for quantization are provided as a resource to the community, freely available and open-source, on Hugging Face Model Hub.

The full-precision model with 316 million parameters is currently among the largest dense transformer encoder models for network biology, on the same order as the natural language dense transformer model BERT-large[4] (336 million parameters, dense) as well as sparse transformer models like natural language model ModernBERT-large[21] (395 million parameters, sparse). As the quantity and diversity of available single-cell transcriptomic data continues to grow, future larger models pretrained on even larger-scale corpuses may open opportunities to achieve meaningful predictions in even more elusive tasks as zero- or few-shot learners. Strong zero-shot performance is particularly valuable in settings without available task-specific data, such as rare diseases and clinically inaccessible tissues, including early human development. Future work will also investigate continual learning strategies to further tune the model toward domains underrepresented in the pretraining corpus, such as cancer. Furthermore, future multimodal models may integrate additional layers of regulation, from

the genome, epigenome, transcriptome, proteome, metabolome and spatial organization with tissues, to gain a more holistic view of the regulatory mechanisms interacting across biological scales to drive cell states.

However, as model size and data grow while available GPU resources remain a limitation, approaches for efficient fine-tuning and inference will be critical to ensure widespread access to models for biological discoveries that have the potential to impact human health. The model quantization method implemented in this work enables significantly higher throughput and cost savings, effectively reducing the compute time for a 30-day large-scale in silico perturbation screen to less than a week, decreasing costs from US$25,000 to <US$5,000, for example. Furthermore, by reducing the memory requirements to 53% and 34% for inference and fine-tuning, respectively, quantization expands the model accessibility to researchers without access to high-memory GPU hardware.

Overall, model quantization enables resource-efficient fine-tuning and inference while preserving biological knowledge, thereby expanding the accessibility of large-scale foundational models for predictions in network biology.

## Methods

### Assembly and rank value encoding of transcriptomes in Genecorpus-104M

**Assembly and uniform processing of single-cell transcriptomes.** We assembled a large-scale pretraining corpus, Genecorpus-104M, comprising ~104 million human single-cell transcriptomes (post-filtering as described below) from a broad range of tissues from 2,903 publicly available datasets (Fig. 1b and Supplementary Table 1). Importantly, DOIs were cross-referenced between all studies to ensure that datasets were unique to avoid inclusion of duplicated cells within the corpus. Of note, there are substantial duplications of datasets across public databases so the total number of unique cells would be highly overestimated if this procedure were not performed (Extended Data Fig. 1a).

Publicly available datasets containing raw counts were collected from National Center for Biotechnology Information (NCBI) Gene Expression Omnibus (GEO), NCBI Sequence Read Archive (SRA), CELLxGENE, Human Cell Atlas, European Molecular Biology Laboratory-European Bioinformatics Institute (EMBL-EBI) Single Cell Expression Atlas, Broad Institute Single Cell Portal, Brotman Baty Institute (BBI)-Allen Single Cell Atlases, Tumor Immune Single-cell Hub (TISCH) (excluding malignant cells), Panglao Database, 10x Genomics, University of California, Santa Cruz Cell Browser, European Genome-phenome Archive, Synapse, Riken, Zenodo, National Institutes of Health (NIH) Figshare Archive, NCBI dbGap, Refine. bio, China National GeneBank Sequence Archive, Mendeley Data, and individual communication with authors of the original studies (Supplementary Table 1). Additional resources for collecting information about suitable studies included Entrez Direct tools and the dataset summary from Svensson et al., Database 2020[22]. Tools utilized in conversion of data to uniform files included loompy, scanpy, anndata, scipy, numpy, pandas, Cell Ranger and LoomExperiment. Gene annotation data were retrieved from Ensembl, NCBI and HGNC (1 November 2023) databases and additionally queried through MyGene[23]. Raw and unfiltered data files were processed to remove empty droplets and debris using STAR version 2.7.8a with the Cell Ranger2.2 (run mode –soloCellFiltered). Datasets were additionally filtered to retain cells that contained a minimum of seven detected Ensembl-annotated protein-coding genes given that the 15% masking used for the pretraining learning objective would not reliably mask a gene in cells with fewer detected genes. Studies were annotated as one or more of the 55 consolidated organs as listed in Fig. 1b.

**Rank value encoding of single-cell transcriptomes.** Each transcriptome was presented to the model as a rank value encoding as

previously described[6]. The rank value encodings are a non-parametric representation of the transcriptome that takes advantage of the many observations of the gene's expression across the entire Genecorpus to prioritize genes that distinguish cell state. Specifically, this method will deprioritize ubiquitously highly expressed housekeeping genes by normalizing them to a lower rank. Conversely, genes such as transcription factors that may be lowly expressed when they are expressed but highly distinguish cell state will move to a higher rank within the encoding. Furthermore, this rank-based approach may be more robust against technical artifacts that may systematically bias the absolute transcript counts value while the overall relative ranking of genes within each cell remains more stable.

The rank value encodings were constructed as previously described[6]. The scaling factor for each gene was derived from the non-zero median value of expression of each detected gene across all cells in the pretraining corpus passing quality filtering that were sequenced on droplet-based platforms, excluding cells with high mutational burdens such as malignant cells and immortalized cell lines. After scaling the expression of each gene, the genes were ordered by the rank of their scaled expression in that specific cell. The rank value encoding for each single-cell transcriptome was then tokenized on the basis of a vocabulary of 20,271 protein-coding genes detected within the pretraining corpus. The vocabulary also included four special tokens: a padding, masking, CLS (classification) and EOS (end of state) token, for a total vocabulary size of 20,275. A CLS and EOS token were added to the beginning and end of each rank value encoding, respectively. The CLS token is standardly abbreviating the term 'classification' as it is used for fine-tuning the model for input-level classification (in this case cell-level classification). In general, however, the CLS token is used here as a global contextual cell representation as it occurs at the beginning every cell with its meaning tuned via the bidirectional attention to the particular cell's context applied during training. The tokenized dataset was stored within the Hugging Face Datasets structure, which is based on the Apache Arrow format that allows processing of large datasets with zero-copy reads without memory constraints.

Of note, this strategy is also space-efficient as the genes are stored as ranked tokens as opposed to the exact transcript values, and we only store genes detected within each cell rather than the full sparse dataset that includes all of the undetected genes. This also prevents wasting computation on zeros, as the model learns from the absence of genes from a rank value encoding without having to explicitly instruct the model that they have zero expression. This is analogous, for example, to how natural language models learn that a statement may have 'positive' meaning based on the absence of 'negative' words, without needing to present the remainder of the absent words from the natural language dictionary at the end of every sentence to explicitly instruct the model they are not present.

### Geneformer architecture, pretraining and quantization
**Geneformer architecture.** Geneformer is composed of dense transformer encoder units, each composed of a self-attention layer and a feed-forward neural network. The original Geneformer model[6] pretrained in June 2021 was an ~10 million parameter model with input size of 2,048. In this work, the input size was expanded to 4,096 genes per cell, which fully represents 93% of the cells in Genecorpus-104M when considering genes within the model vocabulary of 20,271 protein-coding genes. Geneformer models of increasing parameter count were pretrained in this work to evaluate the scaling laws of transcriptional masked learning. The depth of the Geneformer models trained with 38 million, 65 million, 104 million, 228 million and 316 million parameters was 8, 10, 12, 16 and 18 layers, respectively. All models had a width-to-depth aspect ratio of 64, which was optimized by evaluating models trained with the same number of parameters but varying width-to-depth aspect ratios. All models had an embedding-dimension-to-attention-head ratio of 64 and

feed-forward-size-to-embedding-dimension ratio of 4. For example, the 316 million parameter model was 18 layers with 1,152 embedding dimensions, 18 attention heads and a feed-forward size of 4,608. Further parameters are as follows: nonlinear activation function, rectified linear unit; dropout probability for all fully connected layers, 0.02; dropout ratio for attention probabilities, 0.02; standard deviation of the initializer for weight matrices, 0.02; epsilon for layer normalization layers, $1 \times 10^{-12}$. Modeling was implemented in pytorch and using the Hugging Face Transformers library for model configuration, data loading and training.

**Geneformer pretraining and performance optimization.** Geneformer was pretrained with ~104 million (103,877,737) single-cell transcriptomes from Genecorpus-104M excluding cells with high mutational burdens such as malignant cells and immortalized cell lines. Pretraining was accomplished using a masked learning objective, which has been shown in other informational fields[3,4] to improve generalizability of the foundational knowledge learned during pretraining for a wide range of downstream fine-tuning objectives. During pretraining, 15% of the genes within each transcriptome were masked, and the model was trained to predict which gene should be within each masked position in that specific cell state using the context of the remaining unmasked genes. A major strength of this approach is that it is entirely self-supervised and can be accomplished on completely unlabeled data, which allows the inclusion of large amounts of training data without being restricted to samples with accompanying labels. Pretraining hyperparameters were optimized to the following final values: max learning rate, $2 \times 10^{-4}$; learning scheduler, cosine with warm-up; optimizer, Adam with weight decay fix; warm-up ratio, 0.007; weight decay, 0.044; effective batch size (batch size × GPUs), 432 for all models except 228 million parameter model, which had an effective batch size of 216. For the scaling analysis, all models were pretrained for 0.1 epochs. For the full-model pretraining, GF-104M was pretrained for 3 epochs, and GF-316M was pretrained for 1 epoch. Weights & Biases was used for experimentation tracking.

As the input size of 4,096 is considerably large for a full dense self-attention model (for example, BERT[3,4] input size of 512) and transformers have a quadratic memory and time complexity $O(L^2)$ with respect to input size, we implemented distributed GPU training algorithms[24,25] to allow efficient pretraining on the large-scale dataset using Deepspeed. This approach partitions parameters, gradients and optimizer states across available GPUs, offloads processing/memory as possible to a central processing unit to allow more to fit on a GPU, and reduces memory fragmentation by ensuring that long- and short-term memory allocations do not mix. Pretraining was performed in full precision (FP32) and distributed across 8–24 Nvidia H100 80GB GPUs.

**Geneformer quantization and parameter efficient fine-tuning.** Quantization of Geneformer models for fine-tuning was performed using QLoRA[16], a technique that enables parameter- and memory-efficient fine-tuning of large language models by combining low-bit quantization with low-rank adaptation. In this set-up, gradients are propagated through the frozen, quantized base model without updating its weights, and are used to only update the lightweight, task-specific low-rank adapters, enabling memory- and compute-efficient fine-tuning. For both gene-level and cell-level fine-tuning tasks, Geneformer was quantized to 4-bit precision using the NormalFloat4 (NF4) format, enabling fine-tuning on resource constrained hardware without sacrificing model performance.

QLoRA achieves parameter-efficient fine-tuning by integrating two complementary strategies: quantization of the base model and the use of low-rank adapters to perform task-specific updates. Instead of updating all weights in the model, QLoRA freezes the majority of the parameters and introduces trainable low-rank matrices into select layers of the model architecture, typically within the attention mechanisms

(for instance, the key, query, value projection matrices) where the bulk of the computations occur. Low-rank adaptation (LoRA)[26] replaces a full-rank weight update with a low-rank decomposition. For a weight matrix $W$ with shape $N \times N$, which would normally require $N^2$ parameter updates during fine-tuning, LoRA approximates the update $W$ with two matrices of smaller rank: $A$ of shape $N \times r$ and $B$ of shape $r \times N$, where $r$ is much smaller than $N$. This reduces the number of trainable parameters from $N^2$ to $2Nr$, providing a substantial gain in efficiency, particularly when $N$ is large, and $r \ll N$. The low-rank product $\Delta W = AB$ is then added to the original, frozen weight matrix $W$ to preserve the pretrained knowledge while enabling efficient task-specific adaptation.

As the base model weights are frozen, LoRA modules can be trained independently for different tasks and stored separately, which facilitates modular deployment. LoRA's use of low-rank updates is effective because pretrained large language models have low intrinsic dimensionality, meaning there exists a low-dimensional subspace that can be nearly as effective for fine-tuning as optimizing over the full parameter space[27]. Interestingly, larger large language models tend to have lower intrinsic dimensionality because they have already been extensively pretrained on a broad and diverse set of data, allowing them to generalize more effectively with fewer task-specific parameter updates.

To further improve memory efficiency, QLoRA uses quantization techniques to compress the base model before inserting the low-rank adapters. Quantization is the process of reducing the numerical precision of model weights, typically from 16- or 32-bit floating-point values to lower-bit representations such as 8-bit or 4-bit, to reduce memory usage and accelerate computation, while ensuring minimal loss of information from the original weights. QLoRA uses blockwise quantization utilizing the NF4 format (which is more accurate than standard 4 bit floats or integers) that preserves the distributional properties of full-precision weights while enabling substantial compression. For fine-tuning, Geneformer was quantized from 32-bit (FP32) to 4-bit (NF4), also using double quantization, which uses a second nested quantization to save an additional 0.4 bits per parameter.

For inference, the full-precision (FP32) models were quantized to 8-bit precision. The quantization configuration loaded the model in 8-bit precision without using low-rank matrices $A$ and $B$ for fine-tuning. The full-precision weights were directly converted to a lower precision without additional low-rank factorization, such that the original architecture and weights remained intact during the quantization process.

To provide a practical estimate of inference time and cost for in silico perturbation, the GF-V2-316M model was tested either as full precision or quantized in the task of genome-wide in silico deletion for cells with 4,096 genes detected each, which for 30,000 cells results in 122,910,000 total cells processed during inference including both unperturbed and simulated perturbed cells. With the same batch size of 64, genome-wide in silico perturbation for 100 conditions (for instance, cell types or developmental stages) with 300 cells each on 12 Nvidia 80G A100 GPUs would take 32.8 days for the full-precision model but only 5.9 days for the quantized model. With the current price from a low-cost commercial provider of US\$2.56 per Nvidia 80G A100 GPU hour, the experiment would cost US\$24,188 for the full-precision model but only US\$4,361 for the quantized model. Given that the quantization also saves memory, increasing the batch size to utilize the freed memory would result in even greater time and cost benefits. Of note, compute time and cost depends on multiple factors including dataset size, number of genes detected per cell, number of perturbations tested, which model is used, which GPU distribution methods are used, the memory and speed of the GPU resources available, the cost of the specific GPU resources at that given point in time, and so on.

## Masked learning evaluation

Geneformer models with increasing parameters were evaluated on 100,000 held-out cells randomly subsampled from Genecorpus-104M to retain the full diversity of that dataset. For the comparison with baseline masked gene predictions, a Geneformer model of 104 million parameters was utilized, pretraining for 0.1 epochs and evaluating on held-out cells randomly subsampled from Genecorpus-104M. Performance on the masked learning objective increases with pretraining, as shown in the decreasing loss over training time in Fig. 1f; 0.1 epochs was chosen for this evaluation as it represents the compute optimal frontier. The percentage of correct predictions per cell was quantified while masking 15% of the genes. The variable number of masked positions per cell reflects the variable number of genes detected per cell, with the percentage masked always remaining stably at 15%.

The baseline masked gene predictions were derived by identifying the gene most commonly ranked at each position within the rank value encoding in the same 10% of the pretraining corpus observed by the Geneformer model during pretraining. The baseline approach predicts the most likely gene for each rank within the 15% masked genes. This strategy compares the baseline approach for the equivalently challenging task that the Geneformer model is required to perform, namely, identifying which gene within the 20,271 gene vocabulary should be predicted for each rank within the 15% masked genes. This approach also ensures that the distribution for the baseline predictions is not drawn from the evaluation data itself, as some previous studies have done. Furthermore, the evaluation data are of equivalent diversity as the pretraining corpus, ensuring an equitable comparison given that foundation models are tasked with predicting genes within masked positions in the full diversity of tissues, developmental stages and disease states in the pretraining corpus. This ensures that the baseline predictions are not based on a distribution drawn from a narrow evaluation dataset itself, as at the extreme a baseline evaluated on a single cell would be 100% accurate if the evaluation data were used to generate the baseline predictions, which is problematic.

## Contextual gene embedding extraction and robustness to batch-dependent technical artifacts

For each single-cell transcriptome presented to Geneformer, the model embeds each gene into an $N$-dimensional space that encodes the characteristics of the gene specific to the context of that cell. Contextual Geneformer gene embeddings are extracted as the hidden state weights for the $N$ embedding dimensions for each gene within the given single-cell transcriptome evaluated by forward pass through the Geneformer model. Gene embeddings analyzed in this study were extracted from the second-to-last layer of the models as the final layer is known to encompass features more directly related to the learning objective prediction whereas the second-to-last layer is a more generalizable representation. Contextual zero-shot gene embeddings for the quantized versus full-precision models were evaluated for cosine similarity between the same versus different genes within the same or different cell subclasses as annotated within a random subsample of 248,569 normal adult cells within uniformly annotated CELLxGENE data.

To quantify the impact of common batch-dependent technical artifacts on Geneformer gene embeddings, we compared (1) the cosine similarity of embeddings from two randomly selected genes from the same cell (which we expect to have low cosine similarity), (2) the cosine similarity of embeddings from the same gene from two different cells of the same cell type from the same batch (which we expect to have high cosine similarity), and (3) the cosine similarity of embeddings from the same gene from two different cells of the same cell type from different batches (which we expect to have high cosine similarity if the gene embeddings are robust to batch-dependent technical artifacts). We performed the above procedure to quantify (1) platform-related effects using 500 cells iPSCs assayed in parallel on the Drop-seq (single cell) or DroNc-seq (single nucleus) platform[28], (2) preservation-related effects using 330 fresh versus frozen natural killer (NK) cells from the same donor (https://www.10xgenomics.com/datasets/frozen-pbm-cs-donor-a-1-standard-1-1-0 and https://www.10xgenomics.com/datasets/fresh-68-k-pbm-cs-donor-a-1-standard-1-1-0), and (3)

preprocessing-related effects using 4,000 peripheral blood mononuclear cells[29] aligned to genome assembly GRCh37 versus GRCh38 or processed using Cell Ranger versions 2.2.0, 3.1.0 or 7.1.0[30]. Analysis used cell-type annotations provided by original authors for all datasets. Quantification of cosine similarity was performed with $n = 100$ per condition tested (100 cells per batch per cell type). Of note, although we found that Geneformer was robust to the common batch-dependent technical artifacts that we tested, Geneformer was not designed specifically for scRNA-seq batch integration. As such, users may elect to preprocess their limited task-specific data with alternative batch integration methods before using that data for fine-tuning Geneformer toward their downstream task if they find their dataset to be persistently affected by such batch-dependent artifacts.

In addition, separability of the gene embeddings was quantified by the geometric separability index, which was calculated by extracting gene embeddings from a randomly subsampled set of 1,000 cells from the same cell type from the 4,000 PBMCs from analysis 3 above and for genes detected at least 20 times in these cells, measuring whether their nearest neighbor in the embedding space was an embedding of the same gene. The proportion of genes where the nearest neighbor was an embedding of the same gene was 1.0.

## Contextual cell embedding extraction and robustness to batch-dependent technical artifacts

Geneformer cell embeddings, which encode characteristics of the state of that single cell, are generated by extracting the hidden state weights for the $N$ embedding dimensions for the CLS token at the beginning of the given cell's rank value encoding evaluated by forward pass through the Geneformer model. For all applications in this study, we used the second-to-last layer embeddings as discussed above. Contextual zero-shot cell embeddings for the quantized versus full-precision models were evaluated for cosine similarity between the same versus different cell subclasses as annotated within a random subsample of 248,569 normal adult cells within uniformly annotated CELLxGENE data.

To quantify the impact of common batch-dependent technical artifacts on Geneformer cell embeddings, we compared (1) the cosine similarity of embeddings from two randomly selected cells of different cell types (which we expect to have low cosine similarity), (2) the cosine similarity of embeddings from two randomly selected cells of the same cell type from the same batch (which we expect to have high cosine similarity), and (3) the cosine similarity of embeddings from two randomly selected cells of the same cell type from different batches (which we expect to have high cosine similarity if the cell embeddings are robust to batch-dependent technical artifacts). We performed the above procedure to quantify (1) platform-related effects using 500 cells iPSCs assayed in parallel on the Drop-seq (single-cell) or DroNc-seq (single nucleus) platform[28], (2) preservation-related effects using 330 fresh versus frozen NK cells from the same donor (https://www.10xgenomics.com/datasets/frozen-pbm-cs-donor-a-1-standard-1-1-0 and https://www.10xgenomics.com/datasets/fresh-68-k-pbm-cs-donor-a-1-standard-1-1-0), and (3) preprocessing-related effects using 4,000 peripheral blood mononuclear cells[29] aligned to genome assembly GRCh37 versus GRCh38 or processed using Cell Ranger versions 2.2.0, 3.1.0 or 7.1.0 (CR2, 3 or 7, respectively)[30]. We repeated the procedure of quantifying the similarity of cell pairs across the comparisons 1, 2 and 3 above for a total $n = 1,120,000$, 79,200 and 79,200, respectively, for genome reference; $n = 840,000$, 79,200 and 39,600, respectively, for CR2 versus CR3; $n = 1,400,000$, 158,400 and 39,600, respectively, for CR2/3 versus CR7; $n = 40,000$, 20,000 and 19,800, respectively, for the preservation method. Analysis used cell-type annotations provided by original authors for all datasets. In addition, because the preprocessing-related effects allowed comparison for the same exact cell under different conditions, we quantified the impact of the genome reference and Cell Ranger version on the embedding of the exact same cell by cosine similarity, which again we expect to have high

cosine similarity if the cell embeddings are robust to batch-dependent technical artifacts. Of note, the analysis of batch-dependent technical artifacts focused on the same and different cell types of iPSCs and cardiomyocyte subtype 1 as cardiomyocyte subtype 2 annotated by the original authors is both similar and different from cardiomyocyte subtype 1 so could not be clearly categorized as similar or different for the purposes of this batch artifact evaluation.

We also tested the impact of fine-tuning on the batch-dependent technical artifacts related to the lower number of genes detected by single-nuclear versus single-cell sequencing platforms compared with analysis 1 above. GF-104M was fine-tuned to distinguish cell types (as annotated by original authors) using only cells sequenced on the single-cell (Drop-seq) sequencing platform. The fine-tuned embedding space was then evaluated by the same procedure as above. The final embedding layer is known to encompass features more directly related to the learning objective prediction whereas the second-to-last layer is a more generalizable representation. As such, the uniform manifold approximation and projections (UMAPs) reflect the second-to-last layer for the zero-shot model and the last layer for the fine-tuned model as the fine-tuning learning objective was indeed designed to address the separability of cell types as opposed to the general self-supervised pretraining objective. The original data UMAP was generated according to the procedures outlined in the scanpy clustering tutorial, with or without normalization by ComBat[17] or Harmony[18] methods as indicated.

The zero-shot Geneformer embedding space improved the separation by cell type compared with the original data, even after batch correction by ComBat or Harmony, and fine-tuning to distinguish cell type using data from only one platform (single-cell platform) further oriented the embedding space to separate cells by cell type more than by sequencing platform. However, as discussed above, Geneformer was not designed specifically for scRNA-seq batch integration. As such, users may elect to preprocess their limited task-specific data with alternative batch integration methods before using that data for fine-tuning Geneformer toward their downstream task if they find their dataset to be persistently affected by such batch-dependent artifacts.

## Zero-shot, few-shot and fine-tuning evaluation in gene and cell-level tasks

**Gene-level task evaluation.** We tested the full-precision, quantized and alternative models across a diverse panel of biologically meaningful tasks including distinguishing (1) disease genes (dosage-sensitive versus -insensitive transcription factors[31–33]), (2) downstream targets of a transcription factor without perturbation data (*NOTCH1* targets versus non-targets[1,2]), (3) chromatin dynamics from transcriptomic data alone (bivalent versus Lys4-only methylated promoters[34]) and 4) transcription factor regulatory range with no information of genomic distance (transcription factors that act in long versus short range with their targets[35]).

The gene set labels and evaluation data were as previously described[6]. Task 1 is relevant to interpreting copy number variants in genetic diagnosis to determine that genes are sensitive to changes in their dosage. In this analysis, the disease gene task used cells from Genecorpus-30M to ensure that the cells were observed by both the original and current models during pretraining. Of note, it was previously shown that downstream task performance was not impacted by whether the data were included or excluded from the pretraining corpus given that the downstream tasks are distinct from the pretraining task of the masked learning objective[6].

Task 2 is relevant to predicting the direct downstream targets of transcription factors to map gene networks regulating cell states even in the absence of task-specific perturbation data. Task 3 is relevant to predicting chromatin regulation from transcriptomic data alone and distinguishing bivalent marker promoters, which are known to mark key developmental factors in embryonic stem cells and maintain their promoters poised for activation. Performance was evaluated for the 56

highly conserved loci in which bivalent marks were initially described[34] in Fig. 2b and for genome-wide bivalent chromatin marks[36] in Fig. 2c.

Task 4 is relevant to determining the genomic distances over which transcription factor binding influences downstream expression, which is valuable for interpreting regulatory variants and inferring target genes from transcription factor genome occupancy data. Others previously systematically integrated thousands of transcription factor binding and histone modification profiles assayed by ChIP–seq with thousands of gene expression profiles to identify two classes of transcription factor with distinct ranges of regulatory influence. We tested Geneformer's ability to distinguish these long- versus short-range transcription factors using only single-cell transcriptomes from cells undergoing iPSC-to-cardiomyocyte differentiation with no associated ChIP–seq or genomic distance data. This higher-order transcription factor property of regulatory range is a particularly challenging characteristic to infer from transcriptional data alone.

Zero-shot evaluations were performed by training a linear layer on frozen model embeddings to evaluate separability of the zero-shot gene embedding space. Few-shot evaluations were performed by fine-tuning with only 100 example cells. Fine-tuning was performed with 10,000 example cells. All layers were unfrozen to allow weight updates in both the few-shot and fine-tuning evaluations. Alternative models including support vector machines, random forest and logistic regression were fine-tuned as previously described[6]. Geneformer models used a maximum learning rate of 0.0005 for GF-30M and all quantized models and 0.0001 for GF-104M and GF-316M, as larger models often require lower learning rates for stable training. Otherwise no hyperparameter optimization was performed, and all models were trained for 1 epoch with warm-up ratio of 1%, learning rate decay on a cosine schedule and batch size of 1 (standardized for time measurements compared with quantization). QLoRA models were trained with a rank of 128 and alpha of 256. Fine-tuning was tested with 3 different seeds for all models, training on 80% of the genes and predicting on the held-out 20% of genes.

**Cell-level task evaluation.** We tested the separability of the zero-shot versus fine-tuned cell embedding space of full-precision and quantized models by tissue and cell-type attributes and disease states. Zero-shot evaluations were performed by training a linear layer on frozen model embeddings while fine-tuning was allowed to update weights in all layers of the model. Hyperparameter optimization was performed using Ray/HyperOpt for 15 trials for the tissue and cell-type task and 40 trials for the disease task, optimizing learning rate, warm-up ratio, dropout rate, gradient normalization and weight decay for each model. QLoRA models were trained with a rank of 32 and alpha of 64. Performance using the Geneformer embedding space to distinguish tissues and cell types was compared with using scikit-learn's implementation of support vector machines, random forest and logistic regression trained on either raw counts or highly variable genes selected from the training data.

Rather than evaluating each tissue separately, the embedding space was jointly evaluated for separability across all 159 classes of concatenated tissue and cell subclass labels within a random subsample of normal, adult cells within the uniformly annotated CELLxGENE corpus. We used 434,994 cells for training, a separate 144,999 cells were used for validation during hyperparameter tuning and a separate 248,569 cells were held-out in the test set used for the final reported evaluation metrics. For the disease classification, the embedding space was evaluated for separability across 78 classes (77 disease classes, 1 normal class) within a random subsample of cells within the uniformly annotated CELLxGENE corpus. We used 20,000 cells for training, a separate 10,000 cells were used for validation during hyperparameter tuning and a separate 10,000 cells were held-out in the test set used for final reported evaluation metrics. The diseases included 23 cancer types and 54 non-cancer diseases.

## In silico perturbation

We performed in silico perturbation experiments as previously described[6]. In silico deletion was modeled by removing the given gene from the rank value encoding of the given single-cell transcriptome and quantifying the cosine similarity between the original and perturbed gene embeddings of the remaining genes in the single-cell transcriptome to determine which genes were predicted to be most sensitive to in silico deletion of the given gene. Of note, the cosine similarity metric does not directly decode to numerical values of gene expression, but instead represents the model's prediction of the level of impact of the given perturbation on each other gene.

We tested the impact on iPSC-derived cardiomyocytes[19] of in silico deleting *GATA4*, a known congenital heart disease gene, on housekeeping genes[37] compared with GATA4 direct versus indirect target genes. GATA4 direct target genes were defined as any gene significantly dysregulated in response to the *GATA4* variant where GATA4 bound within 20 kb of the gene's transcriptional start site by ChIP–seq in the iPSC disease model[20], whereas indirect target genes were genes significantly dysregulated in response to the *GATA4* variant without GATA4 binding within 20 kb of their transcriptional start site. We tested the impact of in silico perturbation on those genes within the top two quartiles of detections in the cells.

## Reporting summary

Further information on research design is available in the Nature Portfolio Reporting Summary linked to this article.

## Data availability

Genecorpus-104M[38] is available on Hugging Face Dataset Hub at https://huggingface.co/datasets/theodoris-lab/Genecorpus-104M (https://doi.org/10.57967/hf/7859). Source data for Figs. 1–4 and Extended Data Figs. 1–3 are available with this paper.

## Code availability

The pretrained Geneformer models and code for quantization are available on Hugging Face Model Hub at https://huggingface.co/ctheodoris/Geneformer. The repository[39] also includes code for pretraining new models, fine-tuning models for gene- and cell-level tasks, extracting and plotting cell and gene embeddings, performing in silico inhibition and activation, and so on. Detailed documentation is provided at geneformer.readthedocs.io.

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

## Acknowledgements

We thank J. Rae and the Theodoris Lab members for helpful scientific discussions and Argonne National Laboratory for providing GPU resources for experimentation. C.V.T. was supported by grants from the National Institutes of Health (DP5OD036170), Burroughs Wellcome Fund Career Award for Medical Scientists (1022136.01), Cancer Research Institute Technology Impact Award, Biswas Foundation, and Milken Institute. K.P. was supported by NIH/NCI grants R00CA259511 and R37CA295736, the NIH/NCI Cancer Cell Map Initiative, the Parker Institute for Cancer Immunotherapy, funds from the CRISPR Cure for Cancer Initiative, the UCSF Program for Breakthrough Biomedical Research, Cancer Research Institute Technology Impact Award, Lyda Hill Philanthropies, AACR-MPM Oncology Charitable Foundation Transformative Cancer Grant, Colorectal Cancer Alliance, Biswas Foundation and Milken Institute. H.C. was supported by the National Science Foundation Graduate Research Fellowship (2034836). T.N.N. and R.K.M. were supported by the United States Department of Energy Office of Science-Advanced Scientific Computing Research Program (Contract No. DE-AC02-06CH11357).

## Author contributions

H.C. and M.S.V. developed the models and designed and performed the computational analyses. H.C. assembled the pretraining corpus. M.S.V. developed the quantization strategy. J.G.O. contributed to the model pretraining and corpus assembly. S.V.M. contributed to the benchmarking. T.N.N. and R.K.M. advised on distributed training. K.P. and C.V.T. designed the analyses and supervised the work. H.C., M.S.V., K.P. and C.V.T. wrote the paper. All authors edited the paper.

## Competing interests

K.P. is a consultant to Santa Ana Bio, Inc. The other authors declare no competing interests.

## Additional information

**Extended data** is available for this paper at https://doi.org/10.1038/s43588-026-00972-4.

**Correspondence and requests for materials** should be addressed to Christina V. Theodoris.

are available. Primary Handling Editors: Michelle Badri and Kaitlin McCardle, in collaboration with the *Nature Computational Science* team.

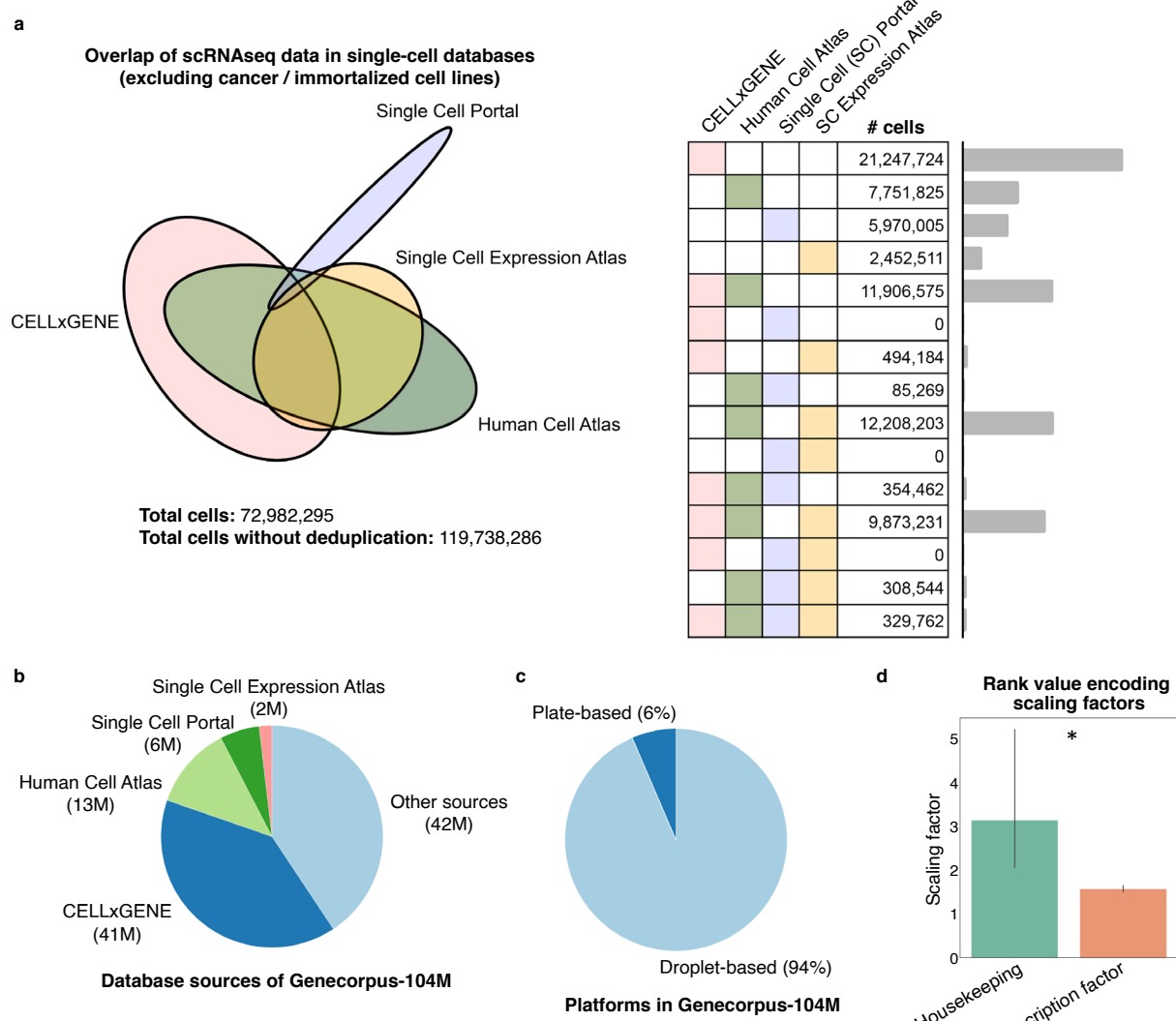

**Extended Data Fig. 1 | Pretraining corpus attributes. a**, Overlap of single-cell transcriptome (scRNAseq) data from single-cell databases (excluding cancer / immortalized cell lines). Without deduplication by DOI, total cells are substantially overestimated. **b**, Database sources of Genecorpus-104M. **c**, Droplet-based vs. plate-based platform composition of Genecorpus-104M. **d**, Transcription factors are normalized by a statistically significantly lower factor (resulting in higher prioritization in the rank value encoding) compared to all genes. Housekeeping genes on average show a trend of a higher normalization factor (resulting in deprioritization in the rank value encoding) compared to all genes (*p = 0.000071 by two-sided Wilcoxon rank sums, FDR-corrected; housekeeping genes n = 11, transcription factors n = 1,384; error bars = standard deviation).

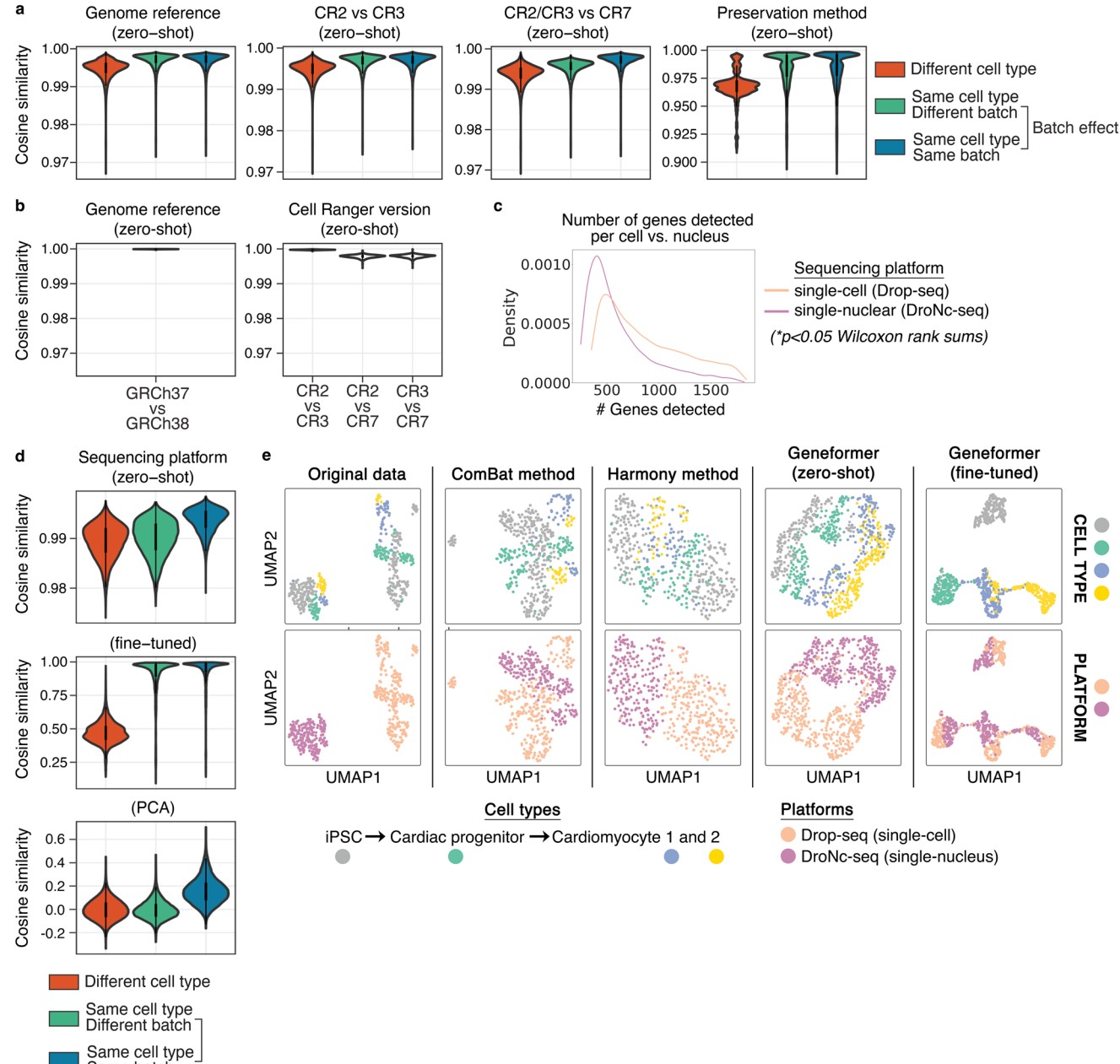

**Extended Data Fig. 2 | Cell embedding robustness to batch effects. a**, Effect on zero-shot Geneformer cell embeddings from batch-dependent artifacts of genome reference (GRCh37 vs. GRCh38), CR version (2 vs. 3 vs. 7), and preservation method (fresh vs. frozen) (cosine similarity of cells from a different cell type vs. cells from the same cell type but a different batch vs. cells from the same cell type vs the same batch: n = 1,120,000, 79,200, 79,200 respectively for genome reference; n = 840,000, 79,200, 39,600 respectively for CR2 vs. CR3; n = 1,400,000, 158,400, 39,600 respectively for CR2/3 vs. CR7; n = 40,000, 20,000, 19,800 respectively for preservation method) (center box limits=upper and lower quartiles). **b**, Cosine similarity of Geneformer cell embeddings for the same cell processed with different genome references (GRCh37 vs. GRCh38) or CR versions (CR2 vs. 3, 2 vs. 7, 3 vs. 7) (n = 800). **c**, Imbalance in the number of genes detected in the same samples[28] split into two portions and sequenced by single-cell vs. single-nucleus RNA-seq. (*p = 0, two-sided Wilcoxon rank sums) **d**, Effect on Geneformer cell embeddings (or PCA of raw expression data) from batch-dependent artifacts of sequencing platform (single-cell vs. single-nucleus

RNA-seq) for either the zero-shot embedding space or the embedding space after fine-tuning the model with only the single-cell RNA-seq data to distinguish cell type (cosine similarity of cells from a different cell type vs. cells from the same cell type but a different batch vs. cells from the same cell type vs the same batch: n = 80,000, 50,000, 49,700 respectively, center box limits=upper and lower quartiles). **e**, UMAP of the original data with or without batch correction with ComBat or Harmony vs. UMAP of the Geneformer zero-shot embedding space or the embedding space after fine-tuning Geneformer with only the single-cell RNA-seq data to distinguish cell type. Of note, the zero-shot Geneformer embedding space improved the separation by cell type compared to the original data, even after batch correction by ComBat or Harmony. Fine-tuning to distinguish cell type using data from only one platform (single-cell RNA-seq) further oriented the embedding space to separate cells by cell type more than by sequencing platform despite the significantly lower number of genes detected per cell in the single-nucleus RNA-seq data.

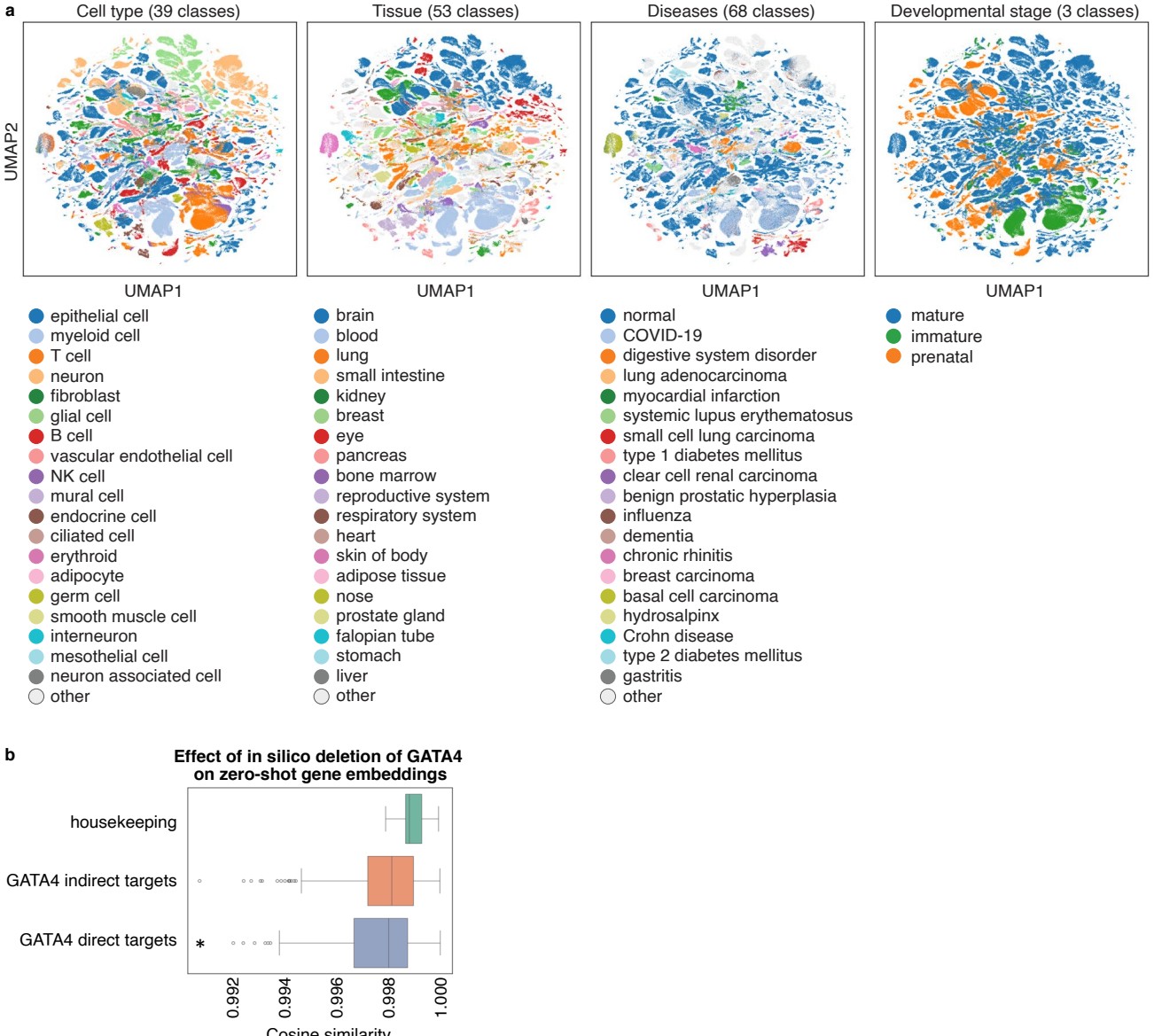

**b**

**Effect of in silico deletion of GATA4
on zero-shot gene embeddings**

**Extended Data Fig. 3 | Zero-shot cell embeddings and in silico perturbation.**
**a**, Zero-shot cell embeddings from GF-316M for 779,905 representative cells
from the CELLxGENE corpus balanced across cell types, tissues, diseases,
and developmental stages and colored by consolidated labels for those cell
attributes. **b**, In silico deletion of *GATA4* in iPSC cardiomyocytes[19] with GF-104M

was significantly more deleterious to previously reported GATA4 direct targets[20]
than to housekeeping genes or GATA4 indirect targets (*p = 0.00023 two-sided
Wilcoxon rank sums; center line=median, box limits=upper and lower quartiles,
whiskers=1.5x interquartile range).

# Reporting Summary

## Statistics

For all statistical analyses, confirm that the following items are present in the figure legend, table legend, main text, or Methods section.

| n/a | Confirmed | |
|---|---|---|
| ☐ | ☒ | The exact sample size (*n*) for each experimental group/condition, given as a discrete number and unit of measurement |
| ☐ | ☒ | A statement on whether measurements were taken from distinct samples or whether the same sample was measured repeatedly |
| ☐ | ☒ | The statistical test(s) used AND whether they are one- or two-sided *Only common tests should be described solely by name; describe more complex techniques in the Methods section.* |
| ☐ | ☒ | A description of all covariates tested |
| ☐ | ☒ | A description of any assumptions or corrections, such as tests of normality and adjustment for multiple comparisons |
| ☐ | ☒ | A full description of the statistical parameters including central tendency (e.g. means) or other basic estimates (e.g. regression coefficient) AND variation (e.g. standard deviation) or associated estimates of uncertainty (e.g. confidence intervals) |
| ☐ | ☒ | For null hypothesis testing, the test statistic (e.g. *F*, *t*, *r*) with confidence intervals, effect sizes, degrees of freedom and *P* value noted *Give P values as exact values whenever suitable.* |
| ☒ | ☐ | For Bayesian analysis, information on the choice of priors and Markov chain Monte Carlo settings |
| ☒ | ☐ | For hierarchical and complex designs, identification of the appropriate level for tests and full reporting of outcomes |
| ☐ | ☒ | Estimates of effect sizes (e.g. Cohen's *d*, Pearson's *r*), indicating how they were calculated |

*Our web collection on statistics for biologists contains articles on many of the points above.*

## Software and code

Policy information about availability of computer code

| Data collection | mygene https://github.com/biothings/mygene.py<br>requests https://github.com/psf/requests |
|---|---|
| Data analysis | The Geneformer models and related code are available on Hugging Face Model Hub (https://huggingface.co/ctheodoris/Geneformer).<br><br>Other software used in this study:<br>anndata https://github.com/scverse/anndata 0.10.7<br>CellRanger https://www.10xgenomics.com/support/software/cell-ranger/latest 2.2, 3, 7<br>cudatoolkit https://github.com/Jimver/cuda-toolkit 11.8.0<br>datasets https://github.com/huggingface/datasets 2.19.2<br>deepspeed https://github.com/microsoft/DeepSpeed 0.15.3<br>hyperopt https://github.com/hyperopt/hyperopt 0.2.7<br>LoomExperiment https://bioconductor.org/packages/release/bioc/html/LoomExperiment.html 1.8.0<br>loompy https://github.com/linnarsson-lab/loompy 3.0.7<br>matplotlib https://github.com/matplotlib/matplotlib 3.9.0<br>multiprocess https://github.com/uqfoundation/multiprocess 0.70.16<br>nccl https://github.com/NVIDIA/nccl 2.21.5.1<br>numpy https://github.com/numpy/numpy 1.26.4<br>optuna https://github.com/optuna/optuna 4.0.0<br>optuna-integration https://github.com/optuna/optuna-integration 4.0.0<br>peft https://github.com/huggingface/peft 0.12.0 |

```
pandas https://github.com/huggingface/peft 2.2.2
pyarrow https://github.com/apache/arrow 15.0.0
python https://www.python.org/ 3.10.13
pytz https://github.com/stub42/pytz 2024.1
pytorch https://github.com/pytorch/pytorch 2.3.1
ray https://github.com/ray-project/ray 2.24.0
scanpy https://github.com/scverse/scanpy 1.10.1
scikit-learn https://github.com/scikit-learn/scikit-learn 1.5.0
scipy https://github.com/scipy/scipy 1.13.1
seaborn https://github.com/scipy/scipy 0.13.2
setuptools https://github.com/pypa/setuptools 69.5.1
STAR https://github.com/alexdobin/STAR 2.7.8a
statsmodels https://github.com/statsmodels/statsmodels 0.14.2
tdigest https://github.com/tdunning/t-digest 0.5.2.2
tokenizers https://github.com/huggingface/tokenizers 0.21.0
tqdm https://github.com/tqdm/tqdm 4.66.4
transformers https://github.com/huggingface/transformers 4.48.2
```

For manuscripts utilizing custom algorithms or software that are central to the research but not yet described in published literature, software must be made available to editors and reviewers. We strongly encourage code deposition in a community repository (e.g. GitHub). See the Nature Portfolio guidelines for submitting code & software for further information.

## Data

Policy information about availability of data

All manuscripts must include a data availability statement. This statement should provide the following information, where applicable:
- Accession codes, unique identifiers, or web links for publicly available datasets
- A description of any restrictions on data availability
- For clinical datasets or third party data, please ensure that the statement adheres to our policy

Genecorpus-104M is available on Hugging Face Dataset Hub at https://huggingface.co/datasets/theodoris-lab/Genecorpus-104M.

## Research involving human participants, their data, or biological material

Policy information about studies with human participants or human data. See also policy information about sex, gender (identity/presentation), and sexual orientation and race, ethnicity and racism.

| | |
|---|---|
| Reporting on sex and gender | N/A this study does not involve human research participants. All data used in this study was assembled from publicly available sources with original studies reporting relevant metadata characteristics. |
| Reporting on race, ethnicity, or other socially relevant groupings | N/A this study does not involve human research participants. All data used in this study was assembled from publicly available sources with original studies reporting relevant metadata characteristics. |
| Population characteristics | N/A this study does not involve human research participants. All data used in this study was assembled from publicly available sources with original studies reporting relevant metadata characteristics. |
| Recruitment | N/A this study does not involve human research participants. All data used in this study was assembled from publicly available sources with original studies reporting relevant study details. |
| Ethics oversight | N/A this study does not involve human research participants. All data used in this study was assembled from publicly available sources with original studies reporting relevant study details. |

Note that full information on the approval of the study protocol must also be provided in the manuscript.

# Field-specific reporting

Please select the one below that is the best fit for your research. If you are not sure, read the appropriate sections before making your selection.

☒ Life sciences    ☐ Behavioural & social sciences    ☐ Ecological, evolutionary & environmental sciences

For a reference copy of the document with all sections, see nature.com/documents/nr-reporting-summary-flat.pdf

# Life sciences study design

All studies must disclose on these points even when the disclosure is negative.

| | |
|---|---|
| Sample size | For the pretraining, we collected ~104 million human single cell transcriptomes from 2,903 publicly available datasets contributing towards a breadth of represented human tissues. These were all datasets we had access to in the public domain at the time of the study and were sufficient based on downstream task evaluations to gain meaningful biological knowledge during the pretraining. |

| | |
|---|---|
| Data exclusions | For the assembly of Genecorpus-104M, human single-cell transcriptomes were gathered from publicly available datasets with cross-referenced DOIs to ensure unique datasets and avoid the inclusion of duplicated cells. Raw and unfiltered data files were processed to remove empty droplets and debris using STAR with CellRanger 2.2 (run mode -soloCellFiltered). Datasets were additionally filtered to retain cells that contained a minimum of seven detected Ensembl-annotated protein-coding genes. |
| Replication | We separated data into training, validation, and test sets for all zero-shot and fine-tuning experiments. Results were replicated across different seeds for the model and different trials for timing; all replication attempts were successful. |
| Randomization | Data was split randomly into training, validation, and test sets for all zero-shot and fine-tuning experiments, either as individual genes for gene classification experiments where the test set contained held-out genes, or as individual cells for cell classification experiments, where the test set contained held-out cells. |
| Blinding | Blinding was not relevant to our study; all results were quantitative and deterministic model outputs. |

# Reporting for specific materials, systems and methods

We require information from authors about some types of materials, experimental systems and methods used in many studies. Here, indicate whether each material, system or method listed is relevant to your study. If you are not sure if a list item applies to your research, read the appropriate section before selecting a response.

## Materials & experimental systems

| n/a | Involved in the study |
|---|---|
| ☒ | ☐ Antibodies |
| ☒ | ☐ Eukaryotic cell lines |
| ☒ | ☐ Palaeontology and archaeology |
| ☒ | ☐ Animals and other organisms |
| ☒ | ☐ Clinical data |
| ☒ | ☐ Dual use research of concern |
| ☒ | ☐ Plants |

## Methods

| n/a | Involved in the study |
|---|---|
| ☒ | ☐ ChIP-seq |
| ☒ | ☐ Flow cytometry |
| ☒ | ☐ MRI-based neuroimaging |

## Plants

| | |
|---|---|
| Seed stocks | *Report on the source of all seed stocks or other plant material used. If applicable, state the seed stock centre and catalogue number. If plant specimens were collected from the field, describe the collection location, date and sampling procedures.* |
| Novel plant genotypes | *Describe the methods by which all novel plant genotypes were produced. This includes those generated by transgenic approaches, gene editing, chemical/radiation-based mutagenesis and hybridization. For transgenic lines, describe the transformation method, the number of independent lines analyzed and the generation upon which experiments were performed. For gene-edited lines, describe the editor used, the endogenous sequence targeted for editing, the targeting guide RNA sequence (if applicable) and how the editor was applied.* |
| Authentication | *Describe any authentication procedures for each seed stock used or novel genotype generated. Describe any experiments used to assess the effect of a mutation and, where applicable, how potential secondary effects (e.g. second site T-DNA insertions, mosaicism, off-target gene editing) were examined.* |

