## [Peer Review File · Nature Computational Science]

Scaling and quantization of large-scale foundation model enables resource-efficient predictions in network biology

Corresponding Author: Dr Christina Theodoris

This manuscript has been previously reviewed at another journal that is not operating a transparent peer review scheme. The manuscript was considered suitable for publication without further review at Nature Computational Science.

Version 0:

Decision Letter:

** Please ensure you delete the link to your author homepage in this e-mail if you wish to forward it to your co-authors. **

Dear Dr Theodoris,

Your manuscript "Quantization of large-scale foundation model enables resource-efficient predictions in network biology" has now been seen by 3 referees, whose comments are appended below. You will see that while they find your work of interest, they have raised points that need to be addressed before we can make a decision on publication.

The referees' reports seem to be quite clear. Naturally, we will need you to address *all* of the points raised.

While we ask you to address all of the points raised, the following points need to be substantially worked on:

- 1) Please provide broader downstream analysis, as requested by Reviewer #1.
- 2) Please provide additional methodological clarifications

Please use the following link to submit your revised manuscript and a point-by-point response to the referees' comments (which should be in a separate document to any cover letter):

Link Redacted

** This url links to your confidential homepage and associated information about manuscripts you may have submitted or be reviewing for us. If you wish to forward this e-mail to co-authors, please delete this link to your homepage first. **

To aid in the review process, we would appreciate it if you could also provide a copy of your manuscript files that indicates your revisions by making use of Track Changes or similar mark-up tools. Please also ensure that all correspondence is marked with your Nature Computational Science reference number in the subject line.

In addition, please make sure to upload a Word Document or LaTeX version of your text, to assist us in the editorial stage.

To improve transparency in authorship, we request that all authors identified as 'corresponding author' on published papers create and link their Open Researcher and Contributor Identifier (ORCID) with their account on the Manuscript Tracking System (MTS), prior to acceptance. ORCID helps the scientific community achieve unambiguous attribution of all scholarly contributions. You can create and link your ORCID from the home page of the MTS by clicking on 'Modify my Springer Nature account'. For more information please visit www.springernature.com/orcid.

We hope to receive your revised paper within three weeks. If you cannot send it within this time, please let us know.

Best regards,

Kaitlin McCardle, PhD
Senior Editor
Nature Computational Science

Reviewers comments:

Reviewer #1 (Remarks to the Author):

This manuscript presents a follow-up study on Geneformer, expanding the pretraining corpus from ~30M to ~104M and ~316M human single-cell transcriptomes. The authors further explore model quantization to improve resource efficiency while retaining biological knowledge. While the study is timely and relevant, several issues in the Results require clarification and further evidence in order to strengthen the work.

1. The authors mention that they excluded cells with high mutational burdens such as malignant cells. More justification is needed for this choice. Excluding malignant cells may limit the model's applicability to cancer biology, which is one of the most critical studied areas of single-cell research.
2. The description of Fig. 2a is insufficiently clear to fully understand how the violin plots were generated. It is not explicitly stated how the reported sample sizes ($n=400, 1600, 3200$) were obtained. It is also unclear whether all possible cell pairs were considered or if random subsampling was performed, and if so, what the sampling strategy was. Moreover, while the text claims that "embeddings of different genes were clearly separated in embedding space," the figure does not visualize the embedding space directly but instead shows distributions of similarity scores, which only provide indirect evidence of separation. I recommend that the authors clarify the sampling procedure, and the link between similarity distributions and the claim of separation in embedding space. Including a direct visualization of the embedding space (e.g., TSNE/UMAP of gene embeddings) would make this result more convincing.
3. The presentation of Fig. 2b requires clarification. 1) All results are reported with only $n=3$ random seeds. Stronger evidence would require more runs. 2) Could the authors provide more explanation for the TF regulatory range prediction task? The GF-104M model outperforms the larger GF-316M model under the few-shot learning setting. Additional clarification or discussion of this result would be helpful.
4. In the cell-level task (Extended Data Fig. 2), the cosine similarity between embeddings of different cell types remains unexpectedly high, raising concerns about the discriminative power of the cell embeddings. Moreover, the specific dataset used for these analyses are not clearly described. Without this information, it is difficult to evaluate the reproducibility and generalizability of the results. The authors should explicitly state which datasets were used, and the sample sizes for each comparison.
5. Could the authors also provide the visualization result for GF-316M, analogous to Fig. 3a, so that readers can directly assess how the largest model represents cell- and tissue-level structure in the embedding space?
6. Again, in Fig. 3b, the few-shot performance of GF-104M appears to surpass that of GF-316M. Could the authors clarify this result? In addition, the performance of other baseline models, SVM and RF, should be included for comparison.
7. While the discussion highlights the improvements of the updated Geneformer over its earlier 10M parameter version, it lacks direct comparisons with other single-cell foundation models (e.g., scGPT, scBERT), which would provide a clearer perspective on relative advances.
8. The downstream analyses presented are somewhat limited. It would strengthen the manuscript if the authors could demonstrate the performance of their quantized models on additional datasets, such as cancer transcriptomes, to assess generalizability to more complex and disease-relevant contexts.

Reviewer #1 (Remarks on code availability):

The 104M version of Geneformer has been uploaded to the Hugging Face Model Hub, together with installation instructions.

Reviewer #2 (Remarks to the Author):

Overall, this is a well-executed, thorough investigation of the transformer-based foundation model paradigm for single-cell RNA sequencing data, applied to a variety of tasks. It is particularly nice to see scaling laws for performance with increased parameters, and interesting to note that quantization does not negatively impact fine-tuning results.

This study would be of interest to the field. However, while the emphasize of the paper on quantization is warranted, the

abstract, discussion, and maybe even title could also be changed to reflect the wider nature of the investigation: scaling, impact of data diversity, and robustness of these models to batch / technical effects are key points of interest, and the paper could benefit from a slight reframing (mainly changes to text).

Comments:

1. Very nice to see scaling laws for a sc-FM model, and improvements due to data diversity.
2. Lines 125-131: Rank based scaling of genes. In the methods, the authors describe the scaling factor as the non-zero median of the gene across the training data, but then say it would prioritize/highlight transcription factors if they important but are still lowly expressed across cells: please clarify the scaling factor and explain how it would change the rank of important genes even if they are lowly expressed.
3. Lines 188-203: Comparison to most likely ranked gene baseline. Discussion of the gene rank baseline is a bit wordy and unclear. Authors could condense it to a single sentence that says they use as a baseline the gene that occurs at a given rank most frequently across the training corpus, and move the longer discussion to the methods.
4. Figure 1h: why did the authors present results for 0.1 epoch in particular? Might be nice to add a supplementary figure showing how the percent of correctly predicted genes increases with training (% correct predictions for 0.01, 0.1, 0.5, 1, epochs?).
5. Figure 2a. Either in this main figure or an extended data, it would be nice to see how the cosine similarity of the actual gene expression values / PCA of the expression values (across cells, to the same dimension as gene embeddings) compares to cosine similarity of model embeddings. This would show how GF is able to remove experimental / technical noise present in the original data.
6. Lines 253-267 and Extended Figure 2d: Cell embeddings. The authors comment that the model is robust to certain technical effects, but struggle to separate different sequencing technologies. In Figure 2d, I am surprised that the cosine similarity between ALL cell embeddings is so high before fine-tuning, and decreases significantly after fine-tuning. I am interested in the cells of the same cell type that show up the lower "tail" of the fine-tuned models in Figure 2d violin plot: are these cells low quality (fewer reads, etc.) or certain cell types that are more varied in natural expression? As I suggested in Figure 2a, it could be nice to show cosine similarity of the expression data after applying PCA to the same dimension as the embedding.
7. Figure 4c: In silico deletion of GATA4. How do these embedding shifts decode to actual changes in gene expression, and do these reflect biology (e.g. if GATA4 deletion should decrease / increase expression, does GF predict that accurately). Also, how do models with fewer parameters perform on this task—does it also improve with model parameters?
8. Fine-tuning GF-104M: In Figures 2b and 3b, after fine-tuning GF-104M performs as well as better than GF-316M. Could the authors comment on this? Although zero-shot performance for GF-316M is better than GF-104M, I would be interested in the authors commenting on the trade-off between pre-training and fine-tuning with increasingly larger datasets (if smaller models can achieve results of larger ones with fine-tuning, how necessary is quantization).

Reviewer #2 (Remarks on code availability):

Code is accessible and well-organized.

Reviewer #3 (Remarks to the Author):

Summary

In this manuscript, Chen et al. demonstrate the effect of model quantization on biological foundational models. They tested as QLoRA as the quantization strategy, and argue it retains key biological information while reducing compute time and costs. To support these claims, they first validated, Geneformer, their previous transformer trained on 30 million cell-specific transcriptomes, scales with model complexity, e.g. parameters and input genes per sample, while using their newly curated 103 million transcriptomes. The authors therefore identified 104 or 316 million parameters are needed to achieve zero-shot learning in 4 biological cases, and applied QLoRA. Using the 316M model, the authors argued there is significant reduction in train/compute time and GPU memory for different fine-tuning tasks. They showed negligible differences between original and quantized cell embeddings and concluded that low rank adapters like QLoRA are effective for optimising biological foundational models.

Overall, the manuscript is significant. It represents a successful quantization attempt to improve our gene regulatory network models. Aside from that, it is worth noting Geneformer has better performance than other transformers in biology and there is opportunity to reach true foundational quality if computational costs can be reduced. This manuscript curates a high volume (100 million) transcriptomes, and judging from the reported zero-shot performance, the authors have perhaps achieved foundational quality that performs fairly well across generalised tasks. A key contribution is the quantization step that enables realistic computing time and cost for data of this scale. However, it needs to improve with clarity on details to

achieve impact before publication. Here are comments may benefit the readership.

Major:

1. Given that GF-104M appears to have best F1 scores compared to GF-316M (Figure 2b, 3b), can the authors comment on why they focused on GF-316M instead? GF-316M has not been shown to discover new biology based on provided data and thus is more likely suffering from overtraining. It seems more of interest to optimise GF-104M.
2. Where do the numbers for computational compute time and costs come from? Can the authors confirm those are specifically measured for Geneformer using a large-scale in silico perturbation screen and is different from training time and costs? While the relative improvement in model training time and memory for QLoRA training are shown, it is unclear what the absolute training time and memory requirements were from the text and the GPU resources utilized (Figure 2d-e). Citing these statistics in the main results with the GPU resources utilised would be helpful to determine if the QLoRA approach makes the 316M parameter model usable to a given reader.
3. Figure 3b shows the F1 scores for tissue and cell type classification. It would be good to differentiate tissue from cell type classification as two different plots as the biology is different. What about disease and developmental stage classifications using quantized cell embeddings?
4. The 104M-gene corpus was not available at the Hugging Face link yet. Besides the model and fine-tuning data, the authors should deposit the scripts and validation data used to recreate the analyses for disease genes, network dynamics, chromatin dynamics, and TF regulatory range for transparency. The reviewer found it challenging to recreate these analyses because it is needed to revisit the Geneformer paper, which further cites the original papers for external links, making it a multi-step process to assess the datasets.

Minor:

1. The article could use some editing for typos and errors and needs to be proofread (e.g. line 551 duplicated 'for's, line 636 'remaining stably'), particularly in methods where (e.g. lines 653-685 is analogous to 687-714)
2. Could the authors comment on $n=3$ for 2d, 2e, 3c, 3d, 4e, and 4f? There is also only 1 visible error bar with negligible values, and the rest are not visible at all (Figure 2d), is that correct?
3. There are several terms used that are very ill-defined, for example, the model is described as learning a 'fundamental understanding' of network dynamics. But it is not defined what is meant by 'fundamental understanding', and it's suggested to tone the language down, for example to something like 'latent representation of gene-gene relationships', which is more in-line with the supporting evidence of the model performance.
4. For figures which plot 'FLOPs', I found this confusing because FLOPs are fixed for a given forward/backward parse of a model with a given parameter size. I believe what the authors mean is 'cumulative FLOPs', which is the cumulative amount of float-point operations performed by the model to achieve the plotted loss on the y-axis. Changing the x-axis to 'cumulative FLOPs' would make this clearer to avoid confusion.
o This comment also extends to figure 1e, where I think the authors are referring to the number of tokens seen by the model.
5. Fig 4A, 'deleterious effect' for the 'in silico' GATA4 perturbation is a stretch, perhaps it is better to state '>GATA4 predicted effect'.

Reviewer #3 (Remarks on code availability):

While my lab has expertise in computational biology and can assess the conceptual advances in this paper, we did not have the ability or expertise to directly test the code. We recommend other groups familiar with these methods provide feedback on the code accuracy and utility.

Version 1:

Decision Letter:

Our ref: NATCOMPUTSCI-25-1819A

3rd February 2026

Dear Dr. Theodoris,

Thank you for submitting your revised manuscript "Scaling and quantization of large-scale foundation model enables resource-efficient predictions in network biology" (NATCOMPUTSCI-25-1819A). It has now been seen by the original referees and their comments are below. The reviewers find that the paper has improved in revision, and therefore we'll be happy in principle to publish it in Nature Computational Science, pending minor revisions to satisfy the referees' final requests and to comply with our editorial and formatting guidelines.

We are now performing detailed checks on your paper and will send you a checklist detailing our editorial and formatting requirements in the next few weeks. Please do not upload the final materials and make any revisions until you receive this additional information from us.

TRANSPARENT PEER REVIEW

Nature Computational Science offers a transparent peer review option for original research manuscripts. We encourage increased transparency in peer review by publishing the reviewer comments, author rebuttal letters and editorial decision letters if the authors agree. Such peer review material is made available as a supplementary peer review file. **Please remember to choose, using the manuscript system, whether or not you want to participate in transparent peer**

review.

Thank you again for your interest in Nature Computational Science. Please do not hesitate to contact me if you have any questions.

Sincerely,

Michelle Badri, PhD

Associate Editor, [Research Cross-Journal Editorial Team](https://www.nature.com/natcomputsci/research-cross-journal-editorial-team) for Nature Computational Science
Nature Portfolio

ORCID

Author names using non-Roman characters

Nature Portfolio journals can support presentation of author names using non-Roman characters in the HTML version of the article. If you wish to, please include author names in parentheses after the Roman-character spelling; [see example online here](https://www.nature.com/articles/s44222-024-00258-2). Currently supported scripts are: Arabic, Chinese, Cyrillic, Devanagari, Greek, Hebrew, Hangul, Japanese and Persian. You will be asked to verify the rendering is correct at proof stage.

Reviewer #1 (Remarks to the Author):

The authors answered my questions well and I have no further concerns.

Reviewer #2 (Remarks to the Author):

The authors have addressed my comments satisfactorily.

Reviewer #2 (Remarks on code availability):

The code available and well-organized, with many recent downloads.

Reviewer #3 (Remarks to the Author):

The authors have sufficiently addressed the reviewer comments and the article is suitable for publication

Version 2:

Decision Letter:

24th February 2026

Dear Dr. Theodoris,

I am delighted to tell you that your manuscript NATCOMPUTSCI-25-1819B has been accepted for publication in Nature Computational Science.

We will be publishing your paper on an accelerated schedule. **Please carefully review the details below and contact us immediately at computationalscience@nature.com if you have any travel plans or other conflicts that may make you unable to respond to us for the next 5-7 days.**

In approximately 2 business days you will receive a link to choose the appropriate publishing options for your paper and complete the appropriate grant of rights necessary to publish your work. As it is vital that this process not be delayed, we strongly encourage you to [check your spam filter and whitelist](https://www.simpleminds.com/how-to-check-your-spam-filter-and-whitelist).

emails/">whitelist the email address do-not-reply@springernature.com to ensure that this message is received.

You will receive a link to your electronic proof via email with a request to make any necessary corrections as soon as possible. You will find that we have made minor changes to enhance the clarity of the text and to ensure that your paper conforms to the journal's style so we ask that you review these proofs carefully to ensure that we have not inadvertently introduced errors or altered the sense of your text in any way.

Please return your proof within 24 hours of receiving it. If you have any questions about your proofs or anticipate any delays please contact rjsproduction@springernature.com immediately.

Once a publication date is set for your paper, the Springer Nature press office will be in touch with the full embargo details. We request that you do not send out your own publicity or contact any journalists until you hear from us that the paper has a confirmed publication date.

If you would like to inform your Public Relations or Press Office about your paper, we suggest that you do so immediately to allow them as much time as possible to prepare an appropriate press release and organize publicity if they choose to do so. Please include your manuscript tracking number NATCOMPUTSCI-25-1819B and the name of the journal, which they will need if they contact our press office.

Authors may need to take specific actions to achieve compliance with funder and institutional open access mandates. If your research is supported by a funder that requires immediate open access (e.g. according to <https://www.springernature.com/gp/open-science/plan-s-compliance> Plan S principles or the <https://www.springernature.com/gp/open-science/us-federal-agency-compliance> NIH public access policy) then you should select the gold OA route, and we will direct you to the compliant route where possible. Because authors warrant under our subscription licensing terms that they haven't committed to licensing any version of their article under a licence inconsistent with the terms of our agreement – including the applicable embargo period – publication under the subscription model isn't suitable for authors whose funders require no embargo.

If you have any questions about our publishing options, costs, Open Access requirements, or our legal forms, please contact ASJournals@springernature.com.

Sincerely,

Michelle Badri, PhD

Associate Editor, <https://www.nature.com/natcomputsci/research-cross-journal-editorial-team>>Research Cross-Journal Editorial Team for Nature Computational Science
Nature Portfolio

P.S. Click here if you would like to recommend Nature Computational Science to your librarian - this will link directly to the Recommend page.

<http://www.nature.com/subscriptions/recommend.html#forms>

** Visit the Springer Nature Editorial and Publishing website at <https://group.springernature.com/gp/group/careers/editorial>>[www.springernature.com/editorial-and-publishing-jobs](https://group.springernature.com/gp/group/careers/editorial) for more information about our career opportunities. If you have any questions please click [here](mailto:editorial.publishing.jobs@springernature.com).**

Reviewer response for:

Scaling and quantization of large-scale foundation model enables resource-efficient predictions in network biology

Firstly, we thank the reviewers for their very helpful comments. We have revised the manuscript to add additional analyses as suggested by the reviewers, particularly to test the models' performance in distinguishing disease states including cancer and quantify the benefit of the largest model in the zero-shot setting.

We believe that these changes greatly strengthened the manuscript and thank the reviewers again for their comments. Below, we address the reviewer comments point by point, with reviewer comments in **black** and our response in **blue**.

Reviewer #1 (Remarks to the Author):

This manuscript presents a follow-up study on Geneformer, expanding the pretraining corpus from ~30M to ~104M and ~316M human single-cell transcriptomes. The authors further explore model quantization to improve resource efficiency while retaining biological knowledge. While the study is timely and relevant, several issues in the Results require clarification and further evidence in order to strengthen the work.

1. The authors mention that they excluded cells with high mutational burdens such as malignant cells. More justification is needed for this choice. Excluding malignant cells may limit the model's applicability to cancer biology, which is one of the most critical studied areas of single-cell research.

We appreciate the reviewer's point and recognize the importance of applications in cancer biology. We exclude cells with high mutational burdens because genes with gain of function mutations may have very different functions than the same genes in other cells, leading to the model learning non-generalizable functions of those genes. Cancer cell lines like K562 have high genomic instability; for example, Zhou et al. found by genome sequencing that only 30% of the genome in K562 cells remained in the diploid state and that their genome harbored hundreds of private protein-altering mutations not found in the general population (424 single nucleotide variants and 148 indels) (Zhou et al., *Genome Research* 2019, PMID: 30737237). Of note, nonmalignant cells from the tumor microenvironment are not excluded from the model's pretraining, as these cells are genomically stable, so the model does learn from these cells within the cancer setting. That being said, as the reviewers note, the exclusion of malignant cells may result in the model having a lower baseline understanding of the gene network rewiring that occurs in malignancy.

To test this, we performed domain-specific continual learning to tune Geneformer to the cancer domain by extending the pretraining with ~14 million cells from cancer studies including matched healthy controls to provide this contrasting context to the model. We also included 1% of the non-cancer cells from Genecorpus-104M to prevent catastrophic forgetting of the general

knowledge of gene network dynamics learned by the model during the initial pretraining. When we tested the model that underwent continual learning with cancer studies in the 78-class disease classification task, we found that it increased zero-shot performance on distinguishing cells from the 23 different cancer types compared to the model checkpoint prior to continual learning, but the difference was not statistically significant. The fine-tuning performance on both cancer and non-cancer and the zero-shot performance on distinguishing the 55 non-cancer disease states (including normal) was similar for both models.

We appreciate the reviewer’s comments and believe that the cancer continual learning strategy warrants in depth investigation with further benchmarking tasks, which are out of scope of the current work. As such, we have added a comment in the discussion about future work to further investigate this approach.

Changes to manuscript:

Added to the discussion:

“Future work will also investigate continual learning strategies to further tune the model towards domains underrepresented in the pretraining corpus, such as cancer.”

2. The description of Fig. 2a is insufficiently clear to fully understand how the violin plots were generated. It is not explicitly stated how the reported sample sizes (n=400, 1600, 3200) were obtained. It is also unclear whether all possible cell pairs were considered or if random subsampling was performed, and if so, what the sampling strategy was. Moreover, while the text claims that “embeddings of different genes were clearly separated in embedding space,” the figure does not visualize the embedding space directly but instead shows distributions of similarity scores, which only provide indirect evidence of separation. I recommend that the authors clarify the sampling procedure, and the link between similarity distributions and the claim of separation in embedding space. Including a direct visualization of the embedding space (e.g., TSNE/UMAP of gene embeddings) would make this result more convincing.

We thank the reviewer for requesting these clarifications. The sample size of 100 cells per condition was selected for each sample size except the genome comparison. We have now updated that analysis to also use 100 cells per condition for consistency. The total n varies between analyses because the various studies have different conditions, such as only two cell types or eight cell types, or in the case of CR2 and CR3, both were separately compared to CR7, resulting in twice as many comparisons. To avoid confusion, we have changed the labels to indicate the n was 100 per condition for simplicity.

Regarding separability of the gene embeddings, embeddings of different genes within the same cell generally have a cosine similarity near 0, while embeddings of the same gene in cells of the same type generally have a cosine similarity near 0.9, indicating the embeddings of the same gene are closer in the embedding space than embeddings of different genes. We further quantified the separability of genes in the embedding space with the geometric separability index (GSI). We extracted gene embeddings from 1000 cells randomly sampled from the same cell type from the 10x PBMC-4K dataset and tested the proportion of genes whose nearest neighbor in the embedding space is an embedding of the same gene, testing genes with at least 20 detections across the 1000 cells. By this metric, a higher score indicates better separability, with the range being 0 to 1. The gene embeddings had a GSI of 1.00, indicating they were separable within the embedding space. (Of note, we recommend quantitative metrics such as GSI or linear probes in the high dimensional space rather than qualitative visual inspection of a compressed two dimensional space, particularly given genes have many attributes that are unlikely to be sufficiently represented in two dimensions.)

Changes to the manuscript:

Methods were clarified for simplicity for sample size used:

“Quantification of cosine similarity was performed with n=100 per condition tested (100 cells per batch per cell type).”

Fig. 2a was updated with n=100 per condition as above for the genome reference (far left) for consistency:

Main text and methods were edited to reflect the GSI metric for quantifying separability of the gene embeddings of different genes:

“We found that embeddings of different genes were clearly separated in embedding space based on the high cosine similarity between embeddings of the same gene and the low cosine similarity between embeddings of different genes, with a high geometric separability index of 1.0. Embeddings of the same gene remained highly cosine similar in cells from different batches, demonstrating robustness to batch effects from the genome reference version, the Cell Ranger version for preprocessing, the cell preservation method, and the sequencing platform, comparing single-cell to single-nucleus RNA-seq (Fig. 2a).”

3. The presentation of Fig. 2b requires clarification. 1) All results are reported with only $n=3$ random seeds. Stronger evidence would require more runs. 2) Could the authors provide more explanation for the TF regulatory range prediction task? The GF-104M model outperforms the larger GF-316M model under the few-shot learning setting. Additional clarification or discussion of this result would be helpful.

We thank the reviewer for this point. Determining the genomic distances over which transcription factor binding influences downstream expression is valuable for interpreting regulatory variants and inferring target genes from transcription factor genome occupancy data. Others previously systematically integrated thousands of transcription factor binding and histone modification profiles assayed by chromatin immunoprecipitation sequencing (ChIP-seq) with thousands of gene expression profiles to identify two classes of transcription factor with distinct ranges of regulatory influence (Chen et al., *Nature Communications* 2020; PMID: 32424124). We tested Geneformer’s ability to distinguish these long- versus short-range transcription factors using only single-cell transcriptomes from cells undergoing iPSC to cardiomyocyte differentiation with no associated ChIP-seq or genomic distance data. This higher-order transcription factor property of regulatory range is a particularly challenging characteristic to infer from transcriptional data alone.

We recognize that the GF-104M model performed slightly better in this task in the few-shot setting. That being said, the GF-316M model outperformed the GF-104M model in this task in the zero-shot setting, with greater gains compared to the difference in few-shot performance. The general trend across all four tasks is that the GF-316M model had stronger overall performance in the zero-shot setting, while both of the larger models performed similarly well in the few-shot setting. Stronger performance in the zero-shot setting is particularly important in settings without available task-specific data, such as rare diseases and clinically inaccessible tissues.

Regarding the number of random seeds tested, 3 random seeds was already sufficiently powered to attain statistical significance in the indicated comparisons. We also demonstrated the comparisons across multiple diverse tasks with a total of 108 Geneformer models evaluated. The analysis shows a consistent pattern of GF-316M improving performance in the zero-shot setting across all tasks.

Changes to manuscript:

We added discussion to further explain the TF regulatory range task and added discussion regarding the value of improving zero-shot learning performance.

“Task (4) is relevant to determining the genomic distances over which transcription factor binding influences downstream expression, which is valuable for interpreting regulatory variants and inferring target genes from transcription factor genome occupancy data. Others previously systematically integrated thousands of transcription factor binding and histone modification profiles assayed by ChIP-seq with thousands of gene expression profiles to identify two classes of transcription factor with distinct ranges of regulatory influence. We tested Geneformer’s ability to distinguish these long- versus short-range transcription factors using only single-cell transcriptomes from cells undergoing iPSC to cardiomyocyte differentiation with no associated ChIP-seq or genomic distance data. This higher-order transcription factor property of regulatory range is a particularly challenging characteristic to infer from transcriptional data alone.”

“Strong zero-shot performance is particularly valuable in settings without available task-specific data, such as rare diseases and clinically inaccessible tissues, including early human development.”

4. In the cell-level task (Extended Data Fig. 2), the cosine similarity between embeddings of different cell types remains unexpectedly high, raising concerns about the discriminative power of the cell embeddings. Moreover, the specific dataset used for these analyses are not clearly described. Without this information, it is difficult to evaluate the reproducibility and generalizability of the results. The authors should explicitly state which datasets were used, and the sample sizes for each comparison.

We thank the reviewer for these important comments. Different embedding spaces may have different absolute cosine similarity value ranges, but if the between-class cosine distance is greater than the within-class cosine distance, then the classes remain separable. Here, the discriminative power of the cell embeddings is demonstrated in the analysis shown in Fig. 3b. The zero-shot test set F1 score per class of near 0.9 indicates that the GF-316M model’s zero-shot cell embedding space has high discriminative power across the 159 cell type / tissue classes.

Regarding the analysis in Extended Data Fig. 2, we ensured that all datasets were referenced, and we added the sample sizes to the methods as well. We thank the reviewer for the clarifying questions.

Changes to the manuscript:

Further specified sample size in the methods:

“We performed the above procedure to quantify A) platform-related effects using 500 cells iPSCs assayed in parallel on the Drop-seq (single-cell) or DroNc-seq (single-nucleus) platform²², B) preservation-related effects using 330 fresh vs. frozen natural killer (NK) cells from the same

donor^{29,30}, and C) pre-processing-related effects using 4000 peripheral blood mononuclear cells³¹ aligned to GRCh37 vs. GRCh38 or processed using Cell Ranger versions 2.2.0, 3.1.0, or 7.1.0³². We repeated the procedure of quantifying the similarity of cell pairs across the comparisons (1), (2), and (3) above for a total n=1,120,000, 79,200, 79,200 respectively for genome reference; n=840,000, 79,200, 39,600 respectively for CR2 vs. CR3; n=1,400,000, 158,400, 39,600 respectively for CR2/3 vs. CR7; n=40,000, 20,000, 19,800 respectively for preservation method.”

5. Could the authors also provide the visualization result for GF-316M, analogous to Fig. 3a, so that readers can directly assess how the largest model represents cell- and tissue-level structure in the embedding space?

We thank the reviewer for this suggestion. We now include the analogous figures for the GF-316M embedding space in the supplementary figure.

Changes to manuscript:

Added supplementary figure with GF-316M embedding space structure:

6. Again, in Fig. 3b, the few-shot performance of GF-104M appears to surpass that of GF-316M. Could the authors clarify this result? In addition, the performance of other baseline models, SVM and RF, should be included for comparison.

We appreciate the reviewer comments. In this case the GF-104M model was able to better discriminate the classes after seeing additional examples in the fine-tuned setting. However, the GF-316M model remains the strongest across all comparisons in the zero-shot setting, which is particularly valuable in settings without available task-specific data. The GF-316M model also surpasses the GF-104M model in both the zero-shot and fine-tuned settings in the newly added disease classification task (Fig. 3c, see reviewer #1 point #8 below).

We have added the SVM and RF comparisons for the cell state classification per the reviewer’s suggestion.

Changes to the manuscript:

We have added the following comment to the discussion:

“Strong zero-shot performance is particularly valuable in settings without available task-specific data, such as rare diseases and clinically inaccessible tissues, including early human development.”

Added SVM and RF to Fig. 3b:

7. While the discussion highlights the improvements of the updated Geneformer over its earlier 10M parameter version, it lacks direct comparisons with other single-cell foundation models (e.g., scGPT, scBERT), which would provide a clearer perspective on relative advances.

We appreciate the reviewer’s point. We focus our comparison of the new model to our original Geneformer model given that third parties have already conducted studies focused on comparing various foundation models including the original Geneformer. For example, in Liu et al, *bioRxiv* 2024 (<https://www.biorxiv.org/content/10.1101/2023.09.08.555192v7.full.pdf>), the original Geneformer achieved a score of 0.85 on cell type annotation (sGPT 0.86, UCE 0.79, scBERT 0.69, SCimilarity 0.32, scFoundation 0.26), a score of 0.91 on gene function prediction

(scGPT 0.78 - others not reported on this plot), and a score of 0.988 on perturbation prediction (scGPT 0.987, SCimilarity 0.986, UCE 0.986, scFoundation took too long to run). Boylan et al, *bioRxiv* 2025 (<https://www.biorxiv.org/content/10.1101/2025.09.22.677811v1.full.pdf>) also recently compared Geneformer to scGPT and GenePT with their method scFME (Single Cell Foundation Models Evaluation) for in silico perturbation evaluation and again found Geneformer significantly outperformed the other models tested.

We also would like to ensure the analyses focus on the primary advances in this work investigating the scaling and quantization of a large-scale foundational model for gene network biology. The quantization and scaling are best compared relative to the same model of different parameter sizes or precisions so that all else is controlled.

8. The downstream analyses presented are somewhat limited. It would strengthen the manuscript if the authors could demonstrate the performance of their quantized models on additional datasets, such as cancer transcriptomes, to assess generalizability to more complex and disease-relevant contexts.

We thank the reviewer for this point. In addition to the gene-level tasks and 159-class cell type and tissue classification task, we now test the quantization within the task of distinguishing cells from 78 different disease classes, 23 of which are cancer diseases. We find no significant differences between the quantized and full precision models in the combined disease classification task.

Changes to manuscript:

Addition of disease classification task in Fig. 3c:

Reviewer #1 (Remarks on code availability):

The 104M version of Geneformer has been uploaded to the Hugging Face Model Hub, together with installation instructions.

We thank the reviewer for reviewing the code availability. The 104M and 316M models are publicly available and open source on the repository.

Reviewer #2 (Remarks to the Author):

Overall, this is a well-executed, thorough investigation of the transformer-based foundation model paradigm for single-cell RNA sequencing data, applied to a variety of tasks. It is particularly nice to see scaling laws for performance with increased parameters, and interesting to note that quantization does not negatively impact fine-tuning results.

This study would be of interest to the field. However, while the emphasize of the paper on quantization is warranted, the abstract, discussion, and maybe even title could also be changed to reflect the wider nature of the investigation: scaling, impact of data diversity, and robustness of these models to batch / technical effects are key points of interest, and the paper could benefit from a slight reframing (mainly changes to text).

We appreciate the reviewer's suggestion and have updated the title to "Scaling and quantization of large-scale foundation model enables resource-efficient predictions in network biology".

Comments:

1. Very nice to see scaling laws for a sc-FM model, and improvements due to data diversity.

We thank the reviewer for acknowledging the value of these investigations.

2. Lines 125-131: Rank based scaling of genes. In the methods, the authors describe the scaling factor as the non-zero median of the gene across the training data, but then say it would prioritize/highlight transcription factors if they important but are still lowly expressed across cells: please clarify the scaling factor and explain how it would change the rank of important genes even if they are lowly expressed.

We thank the reviewer for highlighting this important point. Without the scaling factor, the rank order of the genes would always place the most highly expressed genes at the top of the rank value encoding, thereby prioritizing ubiquitously highly expressed housekeeping genes. However, by scaling the expression by the gene's usual expression range (as represented by the non-zero median across the training data), the genes are ranked by their relative overexpression vs. repression in the given single cell transcriptome being presented to the model. As such, even if a transcription factor is general expressed at a lower range, if it is relatively overexpressed within that range in a given single cell, it will move higher in the rank

value encoding compared to a housekeeping gene that is expressed the same high level as its median (since the scaled value will be >1 for the transcription factor but ~ 1 for the housekeeping gene). We compare the scaling factors in Extended Data Fig. 1d, showing that housekeeping genes have statistically significantly higher scaling factors compared to transcription factors. We have now added further explanation to the text to clarify this point.

Changes to manuscript:

Added further discussion to the text to clarify the rank value encoding method:

“Each cell transcriptome was then presented to the model as a rank value encoding, which is a non-parametric representation of the transcriptome where genes are ranked by their expression in that cell scaled by their non-zero median value of expression across the entire pretraining corpus, as previously described⁶. The scaling factor takes into account each gene’s typical expression range so that genes are ranked by their relative overexpression vs. repression within this range in the given single-cell transcriptome presented to the model. Rather than ranking by absolute expression value, which would generally prioritize ubiquitously highly expressed housekeeping genes, the scaling factor deprioritizes housekeeping genes by scaling them to a lower rank. Conversely, genes such as transcription factors that may be expressed at low levels when they are expressed but have a high dynamic range across distinct cell states will move to a higher rank within the encoding in the cells where they are expressed in the higher end of their given range (Extended Data Fig. 1d).”

3. Lines 188-203: Comparison to most likely ranked gene baseline. Discussion of the gene rank baseline is a bit wordy and unclear. Authors could condense it to a single sentence that says they use as a baseline the gene that occurs at a given rank most frequently across the training corpus, and move the longer discussion to the methods.

We thank the reviewer for this suggestion and have edited the text accordingly.

Changes to the manuscript:

The discussion about the importance of appropriate comparisons was moved to the Methods:

“As such, an appropriate comparison would require that baseline approaches also exactly predict which gene of the 20,271 possibilities should occur in a specific position within evaluation data that matches the diversity of the model’s pretraining corpus rather than limited narrow datasets. Furthermore, the baseline prediction distribution should be drawn from the same diverse pretraining data as the foundation model rather than from a narrow evaluation dataset itself, since at the extreme a baseline evaluated on a single cell would be 100% accurate if the evaluation data is used to generate the baseline predictions.”

4. Figure 1h: why did the authors present results for 0.1 epoch in particular? Might be nice to add a supplementary figure showing how the percent of correctly predicted genes increases with training (% correct predictions for 0.01, 0.1, 0.5, 1, epochs?).

We appreciate the reviewer's question. 0.1 epoch was chosen as it represents the compute-optimal frontier. Because the masked learning objective is the pretraining objective, we can visualize that the loss on this task decreases over the course of the training in Fig. 1f. We added discussion to clarify these points.

Changes to the manuscript:

Added discussion to the masked learning evaluation method:

"Performance on the masked learning objective increases with pretraining, as shown in the decreasing loss over training time in Fig. 1f; 0.1 epoch was chosen for this evaluation as it represents the compute optimal frontier."

5. Figure 2a. Either in this main figure or an extended data, it would be nice to see how the cosine similarity of the actual gene expression values / PCA of the expression values (across cells, to the same dimension as gene embeddings) compares to cosine similarity of model embeddings. This would show how GF is able to remove experimental / technical noise present in the original data.

We appreciate the reviewer's suggestion. While we agree that this would be an interesting analysis to compare the value of the gene embeddings, it may not be possible to practically construct a representation that would be comparable between the two due to the inherent single-dimensional size of gene expression values. The gene embeddings are a learned representation of the genes across many observations of the gene in different contexts. However, the gene expression itself is a single value. As such, comparing different genes within the same cell (far left orange violin in Fig. 2a) would only be able to compare a single digit to another. Across cells, in order to create multiple vectors for each gene, we would need to accumulate the expression across a series of cells of the same number as the model embedding dimensions, and then repeat this for a different set of cells until we had the same number of gene representations as we do from embeddings, but this would necessarily be from a larger number of cells so would not be directly comparable. However, we are able to pursue this analysis for cell embeddings in a directly comparable way, as suggested in point 6 below - please see below for this analysis.

6. Lines 253-267 and Extended Figure 2d: Cell embeddings. The authors comment that the model is robust to certain technical effects, but struggle to separate different sequencing technologies. In Figure 2d, I am surprised that the cosine similarity between ALL cell embeddings is so high before fine-tuning, and decreases significantly after fine-tuning. I am interested in the cells of the same cell type that show up the lower "tail" of the fine-tuned models in Figure 2d violin plot: are these cells low quality (fewer reads, etc.) or certain cell types that are more varied in natural expression? As I suggested in Figure 2a, it could be nice to show cosine similarity of the expression data after applying PCA to the same dimension as the embedding.

We thank the reviewer for these insightful comments. Different embedding spaces have different absolute value ranges, but generally the goal is for different states to be separated within the

embedding space. In this case, the cells of the same type are more cosine similar than the cells of different types, as expected, but the type of sequencing platform used also causes a difference in the cells of the same cell type. The single-nuclear RNA-seq detects a significantly lower number of genes (Extended Data Fig. 2c), causing the cell states to look different than the same states as measured with the single-cell RNA-seq platform. However, when the model is fine-tuned using only one platform, the model is sufficiently oriented to the features important for distinguishing the cell types such that the cells from the other platform are appropriately integrated. It is expected that the fine-tuning causes a greater difference in cosine similarities between different cell types vs. between cells of the same cell type, pushing the cosine similarity of the same cell types closer to 1 and the cosine similarity of different cell types farther away from 1.

The cells that show up in the lower tail of Extended Data Fig. 2d are specifically the two subtypes of cardiomyocytes, annotated by the original authors as cardiomyocyte 1 and 2. Because these two subtypes were included together as one cardiomyocyte type, it led to cells of the same cell type appearing different from one another. To clarify this, we now plot the cosine similarities only using one of the two cardiomyocyte subtypes and discuss this in the methods for clarity.

We also added a plot to Extended Figure 2d of the cosine similarity of the expression data after applying PCA to the same dimension as the Geneformer embeddings, as suggested by the reviewer. In the PCA space, the cells of the same cell type from different batches have a cosine similarity around 0, similarly to cells of different types, indicating a larger batch effect than in the zero-shot Geneformer embeddings, as well as the embeddings after fine-tuning with data from only one platform.

Changes to the manuscript:

We replotted Extended Data Fig. 2d with only one of the two cardiomyocyte subtypes to ensure the categories of same cell type and different cell type are clearer and clarified this in the methods. We also added a plot of cosine similarity of the expression data after applying PCA to the same dimension as the Geneformer embeddings.

Updated Extended Data Fig. 2d:

Added to Methods:

“Of note, the analysis of batch-dependent technical artifacts focused on the same and different cell types of iPSCs and cardiomyocyte subtype 1 as cardiomyocyte subtype 2 annotated by the original authors is both similar and different from cardiomyocyte subtype 1 so could not be clearly categorized as similar or different for the purposes of this batch artifact evaluation.”

7. Figure 4c: In silico deletion of GATA4. How do these embedding shifts decode to actual changes in gene expression, and do these reflect biology (e.g. if GATA4 deletion should decrease / increase expression, does GF predict that accurately). Also, how do models with fewer parameters perform on this task—does it also improve with model parameters?

We thank the reviewer for bringing up this important point. Geneformer represents transcriptomes as nonparametric rank value encodings, and the in silico perturbation effect is measured by the cosine similarity of gene or cell embeddings in the unperturbed vs. simulated perturbed cell. The cosine similarity metric of the gene embeddings does not directly decode to numerical values of gene expression. However, genes that have a greater cosine shift in their embeddings are predicted to be more significantly impacted by the given perturbation. Comparing the 104M and 316M models that both have an input size of 4096, both predict GATA4 direct targets to be the most impacted by GATA4 in silico deletion. The statistical significance of the shift induced by GATA4 deletion on its direct targets vs. housekeeping genes is greater in the 316M model compared to the 104M model (p value 0.000033 for 316M vs. 0.00023 for 104M). The in silico shifts in the 10M model are not directly comparable as the input size of 2048 means that deletion of GATA4 results in a larger proportion of genes affected (1/2048 rather than 1/4096). We added data for the 104M vs. 316M models for this task in Extended Data Fig. 3b.

Changes to the manuscript:

Added data for in silico deletion of GATA4 with the 104M model in Extended Data Fig. 3b:

Added discussion to the Methods regarding the reviewer's point that the cosine similarity measurement does not directly decode to numerical values of gene expression:

"In silico deletion was modeled by removing the given gene from the rank value encoding of the given single-cell transcriptome and quantifying the cosine similarity between the original and perturbed gene embeddings of the remaining genes in the single-cell transcriptome to determine which genes were predicted to be most sensitive to in silico deletion of the given gene. Of note, the cosine similarity metric does not directly decode to numerical values of gene expression, but instead represents the model's prediction of the level of impact of the given perturbation on each other gene.

8. Fine-tuning GF-104M: In Figures 2b and 3b, after fine-tuning GF-104M performs as well as better than GF-316M. Could the authors comment on this? Although zero-shot performance for GF-316M is better than GF-104M, I would be interested in the authors commenting on the trade-off between pre-training and fine-tuning with increasingly larger datasets (if smaller models can achieve results of larger ones with fine-tuning, how necessary is quantization).

We thank the author for this important point. As the reviewer points out, although GF-104M performs slightly better in some cases with additional task-specific examples provided, GF-316M consistently performs best in the zero-shot setting, and the gains here are more significant than the difference between models in the few-shot setting. Because not all settings have task-specific data available, the stronger zero-shot performance of GF-316M is valuable. We have added discussion to the text to highlight this point.

Additionally, the GF-104M model also performed equivalently when quantized, so the quantization still provides benefit for improving resource-efficiency. Of note, the smallest model was not able to achieve the results of the larger models, even with full fine-tuning, so the larger models improve predictions even when task-specific data is available for more challenging tasks.

Changes to the manuscript:

We have added the following comment to the discussion:

“Strong zero-shot performance is particularly valuable in settings without available task-specific data, such as rare diseases and clinically inaccessible tissues, including early human development.”

Reviewer #2 (Remarks on code availability):

Code is accessible and well-organized.

We thank the reviewer for reviewing the code availability.

Reviewer #3 (Remarks to the Author):

Summary

In this manuscript, Chen et al. demonstrate the effect of model quantization on biological foundational models. They tested as QLoRA as the quantization strategy, and argue it retains key biological information while reducing compute time and costs. To support these claims, they first validated, Geneformer, their previous transformer trained on 30 million cell-specific transcriptomes, scales with model complexity, e.g. parameters and input genes per sample, while using their newly curated 103 million transcriptomes. The authors therefore identified 104 or 316 million parameters are needed to achieve zero-shot learning in 4 biological cases, and applied QLoRA. Using the 316M model, the authors argued there is significant reduction in train/compute time and GPU memory for different fine-tuning tasks. They showed negligible differences between original and quantized cell embeddings and concluded that low rank adapters like QLoRA are effective for optimising biological foundational models.

Overall, the manuscript is significant. It represents a successful quantization attempt to improve our gene regulatory network models. Aside from that, it is worth noting Geneformer has better performance than other transformers in biology and there is opportunity to reach true foundational quality if computational costs can be reduced. This manuscript curates a high volume (100 million) transcriptomes, and judging from the reported zero-shot performance, the authors have perhaps achieved foundational quality that performs fairly well across generalised tasks. A key contribution is the quantization step that enables realistic computing time and cost for data of this scale. However, it needs to improve with clarity on details to achieve impact before publication. Here are comments may benefit the readership.

Major:

1. Given that GF-104M appears to have best F1 scores compared to GF-316M (Figure 2b, 3b), can the authors comment on why they focused on GF-316M instead? GF-316M has not been

shown to discover new biology based on provided data and thus is more likely suffering from overtraining. It seems more of interest to optimise GF-104M.

We thank the author for this important point. Although GF-104M performs slightly better in some cases with additional task-specific examples provided, GF-316M consistently performs best in the zero-shot setting, and the gains here are more significant than the difference between models in the few-shot setting. Because not all settings have task-specific data available, the stronger zero-shot performance of GF-316M is valuable. We have added discussion to the text to highlight this point.

Changes to the manuscript:

We have added the following comment to the discussion:

“Strong zero-shot performance is particularly valuable in settings without available task-specific data, such as rare diseases and clinically inaccessible tissues, including early human development.”

2. Where do the numbers for computational compute time and costs come from? Can the authors confirm those are specifically measured for Geneformer using a large-scale in silico perturbation screen and is different from training time and costs? While the relative improvement in model training time and memory for QLora training are shown, it is unclear what the absolute training time and memory requirements were from the text and the GPU resources utilized (Figure 2d-e). Citing these statistics in the main results with the GPU resources utilised would be helpful to determine if the QLora approach makes the 316M parameter model usable to a given reader.

We appreciate the reviewer’s point and agree that providing more specific information about training times on specific resources would be helpful. It is important to note that the absolute values are highly dependent on many factors: the resources available, the dataset size (both numbers of cells and numbers of genes detected in each cell), the training technique used (parallelization, etc), the cost per GPU hour at that given point in time, etc. The numbers cited for compute time and costs are indeed for Geneformer for a large-scale in silico perturbation screen. We now add in the Methods an example calculation for specific resources comparing full precision to quantized models using the same batch size. The true time and cost savings would be even greater if the batch size were increased to utilize the memory freed by quantization, overall leading to a ~\$25,000 experiment costing <\$5,000.

Changes to manuscript:

Added example calculation on example resources to the Methods:

“To provide a practical estimate of inference time and cost for in silico perturbation, the GF-V2-316M model was tested either as full precision or quantized in the task of genome-wide in silico deletion for cells with 4096 genes detected each, which for 30,000 cells results in 122,910,000 total cells processed during inference including both unperturbed and simulated perturbed cells. With the same batch size of 64, genome-wide in silico perturbation for 100

conditions (e.g. cell types or developmental stages) with 300 cells each on 12 Nvidia 80G A100 GPU would take 32.8 days for the full precision model but only 5.9 days for the quantized model. With the current price from a low-cost commercial provider of \$2.56 per Nvidia 80G A100 GPU hour, the experiment would cost \$24,188 for the full precision model but only \$4,361 for the quantized model. Given that the quantization also saves memory, increasing the batch size to utilize the freed memory would result in even greater time and cost benefits. Of note, compute time and cost depends on multiple factors including dataset size, number of genes detected per cell, number of perturbations tested, which model is used, which GPU distribution methods are used, the memory and speed of the GPU resources available, the cost of the specific GPU resources at that given point in time, etc.”

3. Figure 3b shows the F1 scores for tissue and cell type classification. It would be good to differentiate tissue from cell type classification as two different plots as the biology is different. What about disease and developmental stage classifications using quantized cell embeddings?

We thank the reviewer for this valuable comment. For the tissue and cell type, we combined these biological attributes in order to test the model’s ability to distinguish the tissue specificity of cell types that occur across multiple tissues, such as fibroblasts. While distinguishing cell types alone would work as an individual task, combining this with tissue makes the problem more challenging. On the other hand, distinguishing any cell type from different tissues may not be as valid biologically, as B cells from any tissue should be different than endothelial cells, for example, so asking the model to distinguish only on the basis of tissue would lump many disparate cell types together. For the same reason, the developmental stage classification task as a single task may not be as biologically valid, since, similarly, a B cell from any developmental stage should be more different than endothelial cells.

Disease may affect multiple cell types in a coordinated manner, however, so we proceeded to test the models’ ability to classify cells from 78 diseases. The largest model outperformed the smaller models in this task, with the greatest gains again being in the zero-shot setting. There was no significant difference between the quantized vs. full precision models for each parameter size.

Changes to manuscript:

Addition of disease classification task in Fig. 3c:

4. The 104M-genecorpus was not available at the Hugging Face link yet. Besides the model and fine-tuning data, the authors should deposit the scripts and validation data used to recreate the analyses for disease genes, network dynamics, chromatin dynamics, and TF regulatory range for transparency. The reviewer found it challenging to recreate these analyses because it is needed to revisit the Geneformer paper, which further cites the original papers for external links, making it a multi-step process to assess the datasets.

We appreciate the reviewer's comment. The Genecorpus repository currently contains the 30M pretraining corpus as well as the validation datasets for each of the mentioned analyses, including the gene classes and scRNAseq tokenized data, for ease of reproducing the results (https://huggingface.co/datasets/ctheodoris/Genecorpus-30M/tree/main/example_input_files/gene_classification).

Upon acceptance of this manuscript to a peer-reviewed journal, we will similarly deposit the 104M pretraining corpus and the data for these tasks tokenized for use with the V2 models for ease of reproduction. The scripts necessary to recreate these analyses are available as code in the repository with a specific example for gene level analysis in the examples directory (https://huggingface.co/ctheodoris/Geneformer/blob/main/examples/gene_classification.ipynb).

Additionally, we have added references in the methods for each of the datasets for ease of identifying the original data.

Changes to manuscript:

We added the references for the relevant datasets to the methods.

Minor:

1. The article could use some editing for typos and errors and needs to be proofread (e.g. line 551 duplicated 'for's, line 636 'remaining stably'), particularly in methods where (e.g. lines 653-685 is analogous to 687-714)

We appreciate the reviewer pointing this out. We have fixed the typo on line 551. On line 636, we intend to write that this value remains stably at 15%. Lines 653-685 and 687-714 are similar but refer to two different analyses, in one case gene embeddings and in the other case cell embeddings, so we kept these separate since there are a few distinctions that need to be written differently.

Changes to manuscript:

Fixed typo on line 551.

2. Could the authors comment on $n=3$ for 2d, 2e, 3c, 3d, 4e, and 4f? There is also only 1 visible error bar with negligible values, and the rest are not visible at all (Figure 2d), is that correct?

We thank the reviewer for bringing up this point. The values are correct - the time and memory usage was exactly the same across multiple replicates. We have modified the legend to point this out for clarity.

Changes to the manuscript:

Clarified legends as follows:

"($n=3$ trials, $*p<0.05$ by Wilcoxon rank sums, bar=mean, error bars=standard deviation; time and memory usage was equivalent across multiple trials)."

3. There are several terms used that are very ill-defined, for example, the model is described as learning a 'fundamental understanding' of network dynamics. But it is not defined what is meant by 'fundamental understanding', and it's suggested to tone the language down, for example to something like 'latent representation of gene-gene relationships', which is more in-line with the supporting evidence of the model performance.

We appreciate the reviewer's point. We have changed this phrase in the introduction to focus on the application of the model to enable predictions with limited data, as many downstream applications are beyond only gaining the representations themselves. Additionally, we have changed this phrase in the legend of the schematic to indicate that the purpose of the pretraining is for the model to gain foundational knowledge relevant to the domain, removing the term "understanding".

Changes to the manuscript:

Edited the text in the introduction and schematic legend:

"We previously developed a transfer learning strategy for network biology, pretraining a foundational deep learning model, Geneformer, on ~30 million single-cell transcriptomes to enable predictions with limited data in network dynamics⁶."

“Initial self-supervised, large-scale pretraining on a generalizable learning objective yields a pretrained model with foundational knowledge relevant to network dynamics.”

4. For figures which plot ‘FLOPs’, I found this confusing because FLOPs are fixed for a given forward/backward parse of a model with a given parameter size. I believe what the authors mean is ‘cumulative FLOPs’, which is the cumulative amount of float-point operations performed by the model to achieve the plotted loss on the y-axis. Changing the x-axis to ‘cumulative FLOPs’ would make this clearer to avoid confusion.

We thank the reviewer for this suggestion and have modified the label to clarify the x axis refers to compute.

Changes to the manuscript:

Edited labels in Fig. 1f:

o This comment also extends to figure 1e, where I think the authors are referring to the number of tokens seen by the model.

We thank the reviewer for this suggestion and have changed the label for clarity.

Changes to the manuscript:

Edited labels in Fig. 1e:

5. Fig 4A, ‘deleterious effect’ for the ‘in silico’ GATA4 perturbation is a stretch, perhaps it is better to state ‘>GATA4 predicted effect’.

We thank the reviewer for the suggestion and have relabeled the plot accordingly.

Changes to the manuscript:

Fig. 4c-d labels edited:

Reviewer #3 (Remarks on code availability):

While my lab has expertise in computational biology and can assess the conceptual advances in this paper, we did not have the ability or expertise to directly test the code. We recommend other groups familiar with these methods provide feedback on the code accuracy and utility.

We appreciate the reviewer’s comment - the code is open source on the public repository and currently under active use by the research community.